



# A perspective on the next generation of Earth system model scenarios: towards representative emission pathways (REPs)

Malte Meinshausen[1,2], Carl-Friedrich Schleussner[3,4], Kathleen Beyer[5], Greg Bodeker[6], Olivier Boucher[7], Josep G Canadell[8], John S. Daniel[9], Aïda Diongue-Niang[10], Fatimah Driouech[11], Erich Fischer[12], Piers Forster[13], Michael Grose[14], Gerrit Hansen[15], Zeke Hausfather[16,17], Tatiana Ilyina[18], Jarmo S. Kikstra[19,20,21], Joyce Kimutai[22], Andrew King[1], June-Yi Lee[23,24], Chis Lennard[25], Tabea Lissner[26], Alexander Nauels[4,1], Glen P. Peters[27], Anna Pirani[28,29], Gian-Kasper Plattner[30], Hans Pörtner[31,32], Joeri Rogelj[19,20], Maisa Rojas[33], Joyashree Roy[34,35,36], Bjørn H. Samset[27], Benjamin M. Sanderson[27], Roland Séférian[37], Sonia Seneviratne[12], Christopher J. Smith[13,20,38], Sophie Szopa[39], Adelle Thomas[4,40], Diana Urge-Vorsatz[41], Guus J. M. Velders[42,43], Tokuta Yokohata[44], Tilo Ziehn[45], Zebedee Nicholls[1, 2, 16]

[1] School of Geography, Earth and Atmospheric Sciences, The University of Melbourne, Melbourne, Australia
[2] Climate Resource, Melbourne, Victoria, Australia
[3] Integrative Research Institute on Transformations of Human-Environment Systems (IRI THESys) and the Geography Department, Humboldt-Universität zu Berlin, Berlin, Germany
[4] Climate Analytics, Berlin Germany
[5] Climate Futures, School of Geography, Planning and Spatial Sciences, University of Tasmania, Australia
[6] Bodeker Scientific, New Zealand
[7] Institut Pierre-Simon Laplace, Sorbonne Université / CNRS, Paris, France
[8] CSIRO Environment, Canberra, ACT 2601, Australia
[9] NOAA Chemical Sciences Laboratory, USA
[10] Senegal Meteorological Service -ANACIM, Senegal
[11] Mohammed VI Polytechnic University, Ben Guerir, Morocco
[12] Institute for Atmospheric and Climate Science, ETH Zurich, Switzerland
[13] Priestley International Centre for Climate, University of Leeds, Leeds, UK
[14] CSIRO Climate Science Centre, Australia
[15] German Institute for International and Security Affairs (SWP), Berlin, Germany
[16] Stripe Inc., San Francisco, USA
[17] Berkeley Earth, Berkeley, USA
[18] Max Planck Institute for Meteorology, Hamburg, Germany
[19] Centre for Environmental Policy, Imperial College London, London, UK



[20] Energy, Climate and Environment (ECE) Program, International Institute for Applied Systems Analysis (IIASA), Laxenburg, Austria
[21] The Grantham Institute for Climate Change and the Environment, Imperial College London, London, UK
[22] Kenya Meteorological Services, Nairobi, Kenya
[23] Research Center for Climate Sciences, Pusan National University, South Korea
[24] Center for Climate Physics, Institute for Basic Science, Busan, South Korea
[25] Climate System Analysis Group, University of Cape Town, South Africa
[26] The New Institute Foundation, Hamburg, Germany
[27] CICERO Center for International Climate Research, Oslo, Norway
[28] CMCC Euro-Mediterranean Centre for Climate Change, Venice, Italy
[29] Università Cà Foscari, Venice, Italy
[30] Swiss Federal Research Institute WSL, Switzerland
[31] Alfred-Wegener-Institute, Bremerhaven, Germany
[32] Bremen University, Bremen, Germany
[33] Universidad de Chile, Santiago, Chile
[34] Asian Institute of Technology, Thailand
[35] GCP-JU, Jadavpur University, India
[36] CDMR-IIT Guwahati, India
[37] CNRM (Université de Toulouse, Météo-France, CNRS), France
[38] Met Office Hadley Centre, United Kingdom
[39] Laboratoire des Sciences du Climat et de l'Environnement, Université Paris-Saclay, CNRS, CEA, UVSQ, France
[40] University of The Bahamas, Bahamas
[41] Central European University, Vienna, Austria
[42] National Institute for Public Health and the Environment (RIVM), Netherlands
[43] Utrecht University, Netherlands
[44] National Institute for Environmental Studies, Tsukuba, Japan
[45] CSIRO, Aspendale, Australia

*Correspondence to*: Malte Meinshausen (malte.meinshausen@unimelb.edu.au), Carl-Friedrich Schleussner, carl.schleussner@climateanalytics.org

**Abstract.**

In every IPCC Assessment cycle, a multitude of scenarios are assessed, with different scope and emphasis throughout the various Working Group and Special Reports and their respective chapters. Within the reports, the ambition is to integrate knowledge on possible climate futures across the Working Groups and scientific research domains based on a small set of 'framing pathways', such as the so-called RCP pathways from the Fifth IPCC Assessment report (AR5) and the SSP-RCP scenarios in the Sixth Assessment Report (AR6). This perspective, initiated by discussions at the IPCC Bangkok workshop in April 2023 on the "Use of Scenarios in AR6 and Subsequent Assessments", is intended to serve as one of the community contributions to highlight needs for the next generation of framing pathways that is being advanced under the CMIP umbrella for use in the IPCC AR7. Here we suggest a number of policy research objectives that such a set of framing pathways should ideally fulfil, including mitigation needs for meeting the Paris Agreement objectives, the risks associated with carbon removal





strategies, the consequences of delay in enacting that mitigation, guidance for adaptation needs, loss and damage, and for achieving mitigation in the wider context of Societal Development goals. Based on this context we suggest that the next generation of climate scenarios for Earth System Models should evolve towards 'Representative Emission Pathways' (REPs) and suggest key categories for such pathways. These 'framing pathways' should address the most critical mitigation policy and adaptation needs over the next 5-10 years. In our view the most important categories are those relevant in the context of

the Paris Agreement long-term goal, specifically an immediate action (low overshoot) 1.5°C pathway, and a delayed action (high overshoot) 1.5°C pathway. Two other key categories are a pathway category approximately in line with current (as expressed by 2023) near- and long-term policy objectives, and a higher emissions category that is approximately in line with "current policies" (as expressed by 2023). We also argue for the scientific and policy relevance in exploring two 'worlds that could have been'. One of these categories has high emission trajectories well above what is implied by current policies, and

the other has very low emission trajectories that assume that global mitigation action in line with limiting warming to 1.5°C without overshoot had begun in 2015. Finally, we note that timely provision of new scientific information on pathways is critical to inform the development and implementation of climate policy. For the second Global Stocktake under the Paris Agreement in 2028, and to inform subsequent development of Nationally Determined Contributions (NDCs) up to 2040, scientific inputs are required well before 2028. These needs should be carefully considered in the development timeline of

community modelling activities including those under CMIP7.





# 1 Introduction

Having a core set of common pathways to drive Earth Systems Models (ESMs), is essential for the climate science, climate impact and climate policy communities. Such are hereafter referred to as framing pathways since they frame how ESMs can be run with a consistent set of drivers (emissions, concentrations, land surface states, solar activity etc.) to build a range of climate futures which in turn provide a common framing input to impact and vulnerability studies can be conducted (Frieler et al., 2023; Warszawski et al., 2014). The framing pathways thereby can provide a backbone of integration across the IPCC
physical science (Working Group I) and impact (WG II) communities[1] and also to link with socio-economic and mitigation information (WG III) (Figure 1). Other avenues to integrate knowledge, such as global warming levels and cumulative emissions, referred to as additional 'dimensions of integration' in IPCC AR6 (Figure 1.24, IPCC AR6 WG I) (IPCC, 2021), are also important, though the temporal and dynamic dimensions of scenarios are often vital both across biogeophysical and social domains to investigate climate system and impact responses. Earth System Model simulations assessed in the IPCC
AR6 were largely conducted in the process of the Coupled Model Intercomparison Project Phase 6 (CMIP6, Eyring et al 2016), with SSP framing scenarios prepared under a broad community effort and run by ESMs within the ScenarioMIP component of CMIP6 (Riahi et al., 2017; O'Neill et al., 2016; Tebaldi et al., 2021; Meinshausen et al., 2020; Gidden et al., 2019b).

The choice of a core set of framing pathways yields influence well beyond the physical climate science and impact communities. Given its prominence in IPCC reports, as well as in the scientific literature (Riahi et al., 2017), scenario selection
strongly influences the perception of what the climate science and policy communities understand as the range of plausible futures, feeding into climate risk assessments and informing the development of adaptation decisions and assessments of losses and damages, and the assessment of mitigation strategies and ambition. As an example, the inclusion of very high emission scenarios that are well above current policy projections in the core set of scenarios for CMIP6 has led to a continued focus on such a scenario in the literature and climate discourse that has come under criticism for mistaking a worst-case for a business
as usual (Hausfather and Peters, 2020; Huard et al., 2022; Box 3.3 in IPCC AR6 WG3, Riahi et al., 2022). For risk assessments and stress tests, those 'high end emission' pathways might continue to have some relevance as a proxy for 'current policy emission, high end climate response' pathways – in the absence of a systematic exploration of high-end climate sensitivity and carbon cycle feedback of 'current policy' scenarios.

Scenario selection is not just of outstanding importance for climate science (e.g., favouring high signal to noise experiments). Climate science based on scenarios informs climate policy, and society in general, including uses by decision makers, the

---

[1] although that link between Working Groups I and II generally faces some time-lag, as impact studies tend to be based on the previous' cycle ESM outputs - simply as a consequence of how long the research sequence of geophysical ESM output, special domain geophysical impact models, and then economic and ecosystem studies take.





private sector and civil society, with examples ranging across broad areas such as climate litigation, financial risk analysis, and regional adaptation planning (Rajamani et al., 2021; Richters et al., 2022; Otto et al., 2022). Such use cases need to be considered when designing a new framing scenario set. The assessment of scenario-based information is central to the IPCC
in particular to provide climate information that is societally and policy relevant, but not policy prescriptive. To support the IPCC in fulfilling its mandate, we argue that it is important that the scenarios run by ESMs cover a wide range of policy relevant futures. Not considering for example 1.5°C-aligned scenarios would hamper a full information base for decision making (Rogelj et al., 2018). On balance, both high-end and low-end emissions scenarios are needed to explore carbon cycle and climate feedbacks; air pollution control; ecosystem consequences of overshoot - exceedance of and return below level of
global warming; and 'worlds avoided'.

## 1.1  Distinction between 'framing climate pathways', 'socio-economic pathways' and 'scenarios'

The past use of different types of scenarios and pathways within the IPCC's assessment continues to create confusion even among well-informed stakeholders. The general understanding is that the IS92 scenarios (Pepper, 1992; Leggett et al., 1992), SRES scenarios (Nakicenovic et al., 2000), 'Representative Concentration Pathways' (RCPs) and SSPs are all scenarios with
which plausible futures are investigated. While the term scenario is generally used as an overarching term, it is useful here to delineate between 'scenarios' and 'pathways'. Building on the definitional distinction in van Vuuren et al. (2014), we focus here on 'pathways' that describe a climate-related transient evolution of the future (emissions, concentrations, geophysical climate), without any explicit assumptions about socio-economics or policy. In line with this definition, quantified socio-economic futures, derived from qualitative socio-economic narratives, can also be regarded as 'pathways' on their own, as
long as they are not integrated with 'policy pathways', 'emission pathways' or 'concentration pathways' etc. Also following the definitional distinction in van Vuuren et al. (2014), we use 'scenario' to refer to the combination of climate, socio-economic and policy 'pathways' into a coherent and internally consistent plausible future (Box 1)[2]. Note that the IPCC AR6 does not make such a distinction between the definitions of pathways and scenarios.

We call the climate-focused descriptions of plausible futures 'framing pathways' (see white near-half circle in Figure 1). This
naming highlights the relation to the RCP-type sets of emissions, concentrations and other bio-geophysical drivers that are used to drive ESMs. Thus, this perspective on a new generation of pathways focuses on climate pathways only, and not on pathways for socio-economic futures (e.g., population growth, GDP etc.). Separating out these two dimensions has precedent: the RCPs were developed in parallel to the 'socio-economic pathways' (SSPs) (Moss et al., 2010; O'Neill et al., 2014). The motivation for this is that the climate and impact communities can perform their simulations at the same time as the
socioeconomic community determines narratives that are consistent with the climate pathways (RCPs in that case). This so-

---

[2] The IPCC AR6 Glossary, reflecting the scientific literature more broadly, often uses the term pathways and scenarios interchangeably (van Diemen et al., 2022).





called 'parallel process' (Moss et al., 2010) allows the different communities to work in parallel, reducing the time required to generate the outputs, increase their cohesion as well as facilitate their assessment[3].

## 1.2 History and purpose of the 'matrix' approach

The separation of socio-economic and policy assumptions from climate change levels started with the split of the SRES A1

scenario family in 2001 (Nakicenovic et al., 2000). Back then, the high technology-progress storyline of A1 was split up into low, medium and high emission scenarios. From the RCPs onwards, the so-called SSP-RCP matrix (Moss et al., 2010; van Vuuren et al., 2014) was used to explicitly present the climate and socio-economic dimensions as independent dimensions. The different SSPs were represented by one dimension, while the climate outcome was represented by the other. Shared Policy Assumptions (SPAs) were used to vary the climate outcomes (Kriegler et al., 2014). The SSPs themselves were constructed

from two independent axes, challenges to mitigation and challenges to adaptation. The goal of the matrix approach was to allow different communities to assess a similar climate outcome while varying other dimensions (such as international co-operation, global economic growth, equity, adaptation, etc). As some key aspects of the SSP-RCPs are becoming dated, it is now time to take stock of achievements and look for opportunities moving forward (O'Neill et al., 2020; Pirani et al., submitted).

While the SSP-RCPs were the foundation of IPCC AR6 WG I through ScenarioMIP (O'Neill et al., 2016), there was very little assessment of the SSP-RCPs in IPCC AR6 WG III (Riahi et al., 2022). From a mitigation perspective, the SSP-RCP framework used five SSPs across a range of forcing levels and was populated by six integrated assessment models (Riahi et al., 2017; Rogelj et al., 2018). These scenarios were important in IPCC SR15 (Masson-Delmotte et al.), but less so in IPCC AR6 WG III (Shukla et al., 2022) as the scenarios started to become dated, did not explore policy-relevant alternative mitigation strategies

and were superseded by more recent literature. While the original SSP modelling exercise covered all the SSPs more or less evenly (Rogelj et al., 2018; Riahi et al., 2017), the SSP2 'middle of the road' scenario has since been used by most modelling groups as a default socioeconomic pathway and represents more than 90% of the 1202 scenarios with a climate assessment in IPCC AR6 WG III scenarios database (Riahi et al., 2022). The SSP-RCP framework was also focussed around particular challenges (mitigation and adaptation), and had limited scope to address other contemporary questions such as temperature

overshoot (Riahi et al., 2022), equity (ENB, 2023; Kanitkar et al., 2022) or degrowth.

While the climate-focussed framing pathways, the focus of this paper, should largely be treated as separate from these socio-economic dimensions, it is in our view vitally important to assess and explore socio-economic considerations over the coming years, and any scenario framework needs to consider ways to ensure that these aspects can be assessed. In other words, having a common set of geophysical framing pathways needs to be followed by an exploration of multiple socio-economic dimensions

---

[3] As a side note, for the next generation of scenarios after the RCPs, the scenarios used again the SSP storylines, but were, - as a short-hand, and somewhat confusingly - also called SSPx-y scenarios, where the x stands for the socio-economic storyline and y for the RCP-like forcing level by the end of the 21st century. In other words, the socio-economic storylines were kept 'attached' to these key SSPx-y framework scenarios used in WG I, with both advantages and disadvantages.





(adaptation, impact, equity, finance, etc.), potentially also with normative choices. For cases, where the influences of different socio-economic futures is investigated, common practice seems to be that impacts, adaptation, and vulnerability communities use a single climate outcome (like the one from SSP2-4.5) while ex-post varying other assumptions such as population, income, and inequality (rather than exploring SSP1-4.5 or SSP3-4.5, which were not evaluated by ESMs). The exploration of different socio-economic dimensions is however not the bottleneck in physical climate science in the immediate future, as they are not

an input to the ESMs. The bottleneck is the computationally expensive ESMs and hence the need to focus on, and prioritise, a set of a few geophysical framing pathways. Those framing scenarios can be implemented in a coordinated manner by international ESM modelling centres and subsequently, the output of these ESMs can be married with a variety of socio-economic futures to assess vulnerability and impacts and adaptation challenges.

## 1.3  High level framing and history

During past IPCC assessment cycles, scenario expert meetings in 2007  in Noordwijkerhout (Moss et al., 2008) at the end of the AR4, and 2015 in Laxenburg (IPCC, 2016), at the end of AR5, provided recommendations that informed the selection of the key scenarios that were run as part of the 5th and 6th phases of the Coupled Model Intercomparison Project (CMIP5 and CMIP6) run with climate and Earth System Models (ESMs). Specifically, the 2007 Noordwijkerhout IPCC expert meeting decided to extend the lower bound of scenarios towards mitigation scenarios (which was a departure from the 2001 SRES set

of scenarios that only covered non-climate-policy scenarios) by including the so-called RCP3-PD pathway, which can be loosely regarded as a scenario that leads to approximately a 'below 2°C' warming and for the first time considered net negative $CO_2$ emissions in the second half of the 21st century. In March 2015, the IPCC Laxenburg expert meeting again pushed the envelope in order to be policy-relevant, given that much of the policy and impact discussion had shifted to lower warming levels, specifically 1.5°C. Thus, a so-called SSP scenario with a radiative forcing outcome of approximately 1.9 Watt per

square metre by the end of the century was added to the set of key scenarios. This allowed CMIP6 to include Climate Model and ESM runs that were intended to approximately align with the Paris Agreement long-term temperature goal – agreed by the end of 2015 – of pursuing efforts to limit warming to 1.5°C. The IPCC concluded its Sixth Assessment cycle (AR6) with a workshop in Bangkok, April 2023 (Masson-Delmotte et al., 2023) on lessons learned in the AR6 and recommendations for the AR7 and future IPCC assessments of scenarios. Part of the agenda was a breakout group on "recommendations for the scientific

communities involved in modelling", although – unlike at other IPCC scenario workshops - no recommendations were made on the specific point of the future scenario design in terms of forcing or warming levels. The workshop report reflects the discussions held at the meeting, i.e., that "It will be useful to explore differences between high overshoot (C2) and limited overshoot (C1) across WGs, enhancing policy relevance."  The need to inform the second Global Stocktake under the Paris Agreement in 2027-2028 was also highlighted (Masson-Delmotte et al., 2023). The workshop called on the larger scientific

community to provide input, which is the motivation for this paper.





**Box 1 - Definitional use of pathways and scenarios**

This box attempts to add nuance to some earlier definitions of 'scenarios' and 'pathways' as used in this paper. It is acknowledged that multiple overlapping definitions with different foci exist for scenarios and pathways, and the definitional use adopted here is meant for clarity of the presented concepts, not as a proposal for an overarching (re-)definition. Our use of the term is adapted from the glossary definitions in IPCC AR6 (van Diemen et al., 2022), which generally have a broader scope. See also van Vuuren et al. (2014) for a useful distinction between scenarios and pathways, which we largely follow here.

**'Pathways':** Pathways quantitatively describe the temporal evolution of natural and/or human systems towards a future state, such as emission, concentration, radiative forcing, warming, techno-economic, and/or socio-behavioural trajectories.

**'Scenario':** We follow here the IPCC Glossary definition, which is "A plausible description of how the future may develop based on a coherent and internally consistent set of assumptions about key driving forces (e.g., rate of technological change, prices) and relationships. Note that scenarios are neither predictions nor forecasts but are used to provide a view of the implications of developments and actions." (van Diemen et al., 2022). In addition to the IPCC definition, we refer to a 'scenario' in this paper as the sum of coherent, internally consistent, and interdependent pathways, each representing a domain (emission, concentration, warming, hazards, socio-economic and/or policy). A scenario of interest here does not have to have a complete representation across all the domains mentioned above or across all three 'IPCC Working Group' domains but can stretch across more than just the physical climate or socio-economic domains.

**'Framing climate pathways':** Framing climate pathways in AR5 and AR6 were the RCP components of the SSP-RCP framework. While the name suggested that these were 'concentration' pathways only, they were actually a set of physical climate pathways from emissions (used for some emission-driven ESM runs) and corresponding concentrations (used for the concentration-driven ESM runs). We also include the 'assessed' global warming and ESM outcomes of these pathways into the broad definitions of 'framing climate pathways'. Thus, 'framing climate pathways' are a collection of internally consistent pathways across emissions, concentrations, radiative forcing, global-mean temperatures, and geophysical hazards (e.g., changes in extreme rainfall), i.e., pathways from the geophysical climate science or IPCC WG I domain. Those pathways can be anchored predominantly in a set of emission pathways, so that concentrations, projected global-mean warming, and regional climate characteristics are either consistent best-estimated pathways related to those emission pathways, or a (probabilistic) range of outcomes that is consistent with those emission pathways, reflective of the current state of uncertainties. In other words, 'framing climate pathways' is the overarching term used here for REPs (representative emission pathways), RCPs (representative concentration pathways), RWPs (representative warming pathways) etc, which





includes not only the emissions in the case of REPs, but then also the corresponding (range of) concentrations, warming outcomes etc. Framing climate pathways span the geophysical climate science domain (see white underlay in Figure 1).

**'Representative Emission Pathways':** Emission pathways are a set of emission trajectories of greenhouse gases, aerosols and their precursors, and land-use, i.e., a collection of (human-induced) driving forcers on the climate on the basis of which ESMs, Earth System Models of Intermediate Complexity (EMICs) or calibrated reduced complexity models (emulators) which are able to produce concentrations and geophysical climate projections. The representative emission pathways presented here were chosen for being a representative subset to align with the science, policy, and other objectives outlined in this paper. If concentration or warming pathways are chosen that are consistent with the REPs, then they would primarily be referred to as REPs. Such a usage implies that the label 'REP' can be used for the 'framing climate pathway' in its entirety, rather than only for the set of emission trajectories. This dual narrow (for the emissions only) and broad (for all geophysical climate domains) use of the term REPs is similar to the historical use of RCPs, with the label 'RCP' often used to refer to for either just the concentration trajectories, or also the corresponding emissions, warming and geophysical hazard outcomes.

**'Representative X pathways':** A representative subset can be selected not only in the emission sub-domain, but also for concentrations, global-mean surface temperatures or other hazard variables ('X' then standing for 'concentrations', 'warming', 'hazard' or similar), or even socio-economic variables (see Figure 1). They can, but do not have to be, internally consistent with the 'representative emission pathways' and can serve different purposes.

## 1.4  Towards Representative Emission Pathways (REPs)

In this perspective we identify categories that could inform the development of framing climate pathways for ESMs based on expert and stakeholder discussions across various communities. Those discussions started during the AR6 cycle and continue with the open participation and review approach used to develop this paper and will be held in multiple other fora as well. As a tentative name, we suggest 'Representative Emission Pathways' or REPs. The basis is a close alignment with the RCPs of CMIP5 and the RCP part of the SSP-RCP matrix. Future ESM runs may be predominantly emission driven (for at least carbon

emissions) to capture carbon cycle uncertainties, therefore we suggest changing the term from 'concentration' to 'emissions', so that the new generation is called 'Representative Emission Pathways' rather than 'Representative Concentration Pathways'[4]. The shift to an emissions-driven framing encourages exploration of additional degrees of freedom in scenario definition:

---

[4] While ideally non-$CO_2$ forcers will also be included at the point of emissions and precursors, future ESMs runs will continue to implement some of those in terms of concentration or abundance inputs ($N_2O$, HFCs, PFCs, ODS, aerosols) in acknowledgement both of ESM capacities and computational efficiency.





regional aerosol emissions, land use strategy and carbon removal where process representation in ESMs can add to understanding (Sanderson et al., in prep.), but it does not restrict the ability to perform concentration-driven runs.

To enable the strong participation of the full set of ESMs, including those that are not yet capable of being emission-driven for $CO_2$, $CH_4$, $N_2O$ and to incorporate the effect of many other climate forcers for which reduced complexity models remain a more efficient choice in terms of computing time, 'best-estimate' concentration forcings would be available to accompany the REPs. The 'best-estimate' projection of concentrations could be a reflection of the 'best-estimate' expert judgement in the most recent IPCC assessment or a more recent expert judgement (such as including new insights, e.g., natural $CH_4$ emission

dynamics in wetlands, Kleinen et al., (2021)) and could be produced by calibrated emulators (Cross-Chapter Box 7.1, IPCC AR6 WG I, Forster et al. (2021)). The input data provision for ESMs could even be extended by providing high and low concentration projections for the REPs, so that even ESMs that cannot start from $CO_2$, $CH_4$ or $N_2O$ emissions could include the gas cycle uncertainty to some extent, if desired.

While REPs will most likely be derived from Integrated Assessment Model (IAMs), we argue that the e REPs should remain

separated from the underlying socio-economic scenarios as previously done under the RCPs (Moss et al., 2010). We recommend that such a separation is essential to enhance uptake and facilitate exploration of alternative socio-economic and other dimensions by adaptation, equity, finance and other scientific communities outside the geophysical science realm.

## 1.5 REPs are only a small part of the overall scenario spectrum

We reiterate that these framing climate pathways are only a small subset of the scenarios that play an important role in IPCC

assessments. Working Group III investigates thousands of socio-economic pathways from the independently generated scientific literature, including their sectoral, national, regional and socio-economic dimensions. And even in AR6, the SSPx-y scenarios did not feature prominently in WG III, where it was rather the so-called Illustrative Mitigation Pathways (IMPs) that highlighted choices about technology, infrastructure and behavioural responses. As frequently pointed out in the IPCC approval sessions, it is also important to consider the equity dimension of scenarios. These points imply a renewed effort to

cross-examine each of the future climate change levels under a broad range of socio-economic futures. At this stage, while developing REPs as input forcings for ESMs, it is not necessary to finalise or pre-empt the scope of socio-economic futures and national level scenarios, given the importance of exploring a greater diversity thereof. A task of future IPCC reports will be to reflect the whole spectrum of research on socio-economic futures and national scenarios. The need to coalesce on a set of framing climate pathways for the ESMs only arises due to the ESMs' multi-year lead times and high computational

costs(Moss et al., 2010). But even for ESMs, a set of framing scenarios does not pre-empt a number of additional investigations, e.g., to allow for both individual modelling groups to pursue their own scenarios or new intercomparison projects to be added later, like ZECMIP in the CMIP6 cycle (Jones et al., 2019; MacDougall et al., 2020). We do not expect the REPs to be the only reflection of scenario issues in the wake of the just concluded IPCC AR6 - with the ultimate aspiration being that the next cycle - AR7 will have a broad range of suitable ESM scenarios to pick from - from a policy-relevance, climate services and

scientific point of view (see e.g. Pirani et al., submitted).



**Figure 1 – Our conceptual overview of climate change scenarios, representative emission pathways (REPs) and the framework climate pathways.** Scenarios are constructed from socio-economic, emission, concentration and impact pathways that are internally





consistent with each other ('+' in middle). We propose to anchor the input for Earth System Models around 'Representative Emission Pathways' (REPs) ('1'), in contrast to anchoring around RCPs as done in IPCC AR5 and IPCC AR6. However, the intention is to flexibly allow different starting points for different communities. For the gas-cycles that are not represented in some Earth System Models, reduced complexity models can translate emission pathways to concentration pathways. The 'best-estimate' ('2') concentration projections in line with REPs can be labelled RCPs, although the 'representative pathways' across different subdomains (emissions, concentrations, socio-

economic) do not necessarily have to be consistent. Furthermore, different models and approaches would translate emissions to concentrations differently, spanning an uncertainty range ('3'). 'Representative Warming Pathways' (RWPs) can provide another entry point into scenario design, closely aligned by a consideration of 'Global Warming Levels' (GWLs) ('5'). If derived from REPs, the range of derived warming pathways would represent both gas-cycle and climate feedback uncertainties ('4'). Geophysical hazard models could then be driven by the output from ESMs to produce hazard pathways. Potentially, one could also design Representative Hazard Pathways (RHPs)

either as best-estimate representations of REP's hazards, or independently. The segment of the cause-effect chain from emissions to hazards, computed by a chain of climate and other numerical geophysical models, is here called the 'Framework Climate Pathways' (bold arrow on right side on white circle segment and '7'). Taking into account adaptation options, potentially as 'Representative Adaptation pathways' (RAPs), the so-called RIPs ('Representative Impact and risks Pathways') could be derived, taking into account vulnerability and exposure, with a full uncertainty propagation from REPs now spanning a wide range for each starting REP. Another entrance point into scenario design

are socio-economic narratives, which can be translated into quantitative socio-economic pathways. In this terminology, the quantitative shared socio-economic pathways SSPs (O'Neill et al., 2014; Riahi et al., 2017) would be termed 'Representative Socio-economic Pathways' (RSPs). Those socio-economic pathways would ideally take into account some climate impacts (dashed line, '8'), an under-represented feedback so far. In combination with mitigation policy assumptions (such as the 'Shared Policy Assumptions', Kriegler et al. (2014)), which are here coined 'Representative Transition Pathways', integrated assessment models can derive a large set of emission pathways ('9') to

investigate mitigation options. Even within this diagram and current processes, there are still factors such as socio-economic inertia and psychosocial delays in implementation of climate mitigation and adaptation actions that are not currently considered.

In the subsequent sections, we first lay out a few design criteria and needs from a policy point of view that, in our opinion, geophysical framing scenarios should ideally meet. Subsequently, we provide an overview from a scientific point of view of

design criteria that geophysical framing scenarios should meet. We then identify pathway categories that could inform the design of REPs to meet these policy and science objectives within the constraints of a potential scenario ensemble CMIP7 design.

## 2   Policy context for framing pathways

The decisions of the United Nations Framework Convention on Climate Change (UNFCCC) and the Paris Agreement and

subsequent decisions are central to the policy context. The new set of framing climate pathways would need to meet the related key mitigation (Sec. 2.1), and adaptation and loss and damage information needs (Sec. 2.2), by addressing, to the extent possible, important policy-relevant questions collated below in a non-comprehensive list (Sec. 2.3).

### 2.1  Focus on Paris Agreement relevant scenarios from mitigation decision maker viewpoint.

The elements outlined in the Paris Agreement, more specifically in relation to the long-term temperature goal expressed in

Article 2.1, and Article 4.1 should be fully explored (Schleussner et al., 2022). Under the UNFCCC, the second Periodic Review of the long-term global goal under the Convention and of overall progress towards achieving it has just concluded its work in 2022. Also, the global stocktake provides a checkpoint on whether aggregate emission levels by Parties are consistent with the long-term goals (UNFCCC, 2022a). The Conference of the Parties under the UNFCCC explicitly "acknowledges that





limiting the global average temperature increase to 1.5°C above pre-industrial levels with no or limited overshoot would avoid
increasingly severe climate change impacts" (UNFCCC, 2022b), building upon the conclusions of the IPCC special report on
1.5°C (Hoegh-Guldberg et al., 2018), as well as the IPCC AR6 WG I and WG II reports (Masson-Delmotte et al., 2021; Pörtner
et al., 2022). They also concluded that while "information and knowledge have improved significantly since the first periodic
review (2013–2015)" (UNFCCC, 2022b, paragraph 7), "there continue to be important information and knowledge gaps" in
relation to its scope, including on "the long-term global goal and scenarios towards achieving it in the light of the ultimate
objective of the Convention". The scientific community is explicitly encouraged to address those gaps. We are of the view that
any scenario design process needs to be very cognisant of this explicitly expressed call by governments for more scientific
information on Paris Agreement compatible scenarios.

Against a backdrop of 'the emission world avoided' context provided by high-end emission scenarios, it is paramount for
decision makers to also understand the implications of stronger mitigation efforts in terms of climate benefits and avoided
impacts. Whether we follow a scenario that delays mitigation efforts by 10, 20 or 30 years and reaches net-zero $CO_2$ emissions
by 2050 or 2060 or 2070 makes trillion-dollar differences in terms of directing government incentives and private capital
(Riahi et al., 2022; van der Wijst et al., 2023), but also in terms of adaptation costs, limits to adaptation, irreversible loss and
economic and non-economic costs of anticipated losses and damages (Pörtner et al., 2022). While natural variability in any
single year influences global-mean temperatures by ±0.25°C (Box 4.1 in IPCC AR6 WGI, i.e., Lee et al., 2021), climate
extremes (Seneviratne et al., 2021) and impacts that reflect long-term, cumulative climate changes (e.g. glacier melt or sea
level rise) can be substantially different between a scenario peaking at 1.6°C or 1.8°C in the middle of the century (Mengel et
al., 2018; Pfleiderer et al., 2018). Only immediate action will slow-down anthropogenic warming in the near-term (McKenna
et al., 2021), a crucial element to enable sustainable development (Schleussner et al., 2021).

## 2.2 Comprehensive range to inform risk management, adaptation needs and loss and damage assessments.

Output from ESMs based on the framing pathways should explore a comprehensive range of plausible warming futures.
Plausible very high end / worst case outcomes are a key foundation for risk management andadaptation decision making as
well as on loss and damage. Also, those high-end outcomes can be indicative of the high-end tail of the distributions, including
high-end climate sensitivity, that are helpful in examining limits to adaptation and projected levels of loss and damage that
may need to be addressed. Information about high-end global warming outcomes is of particular importance on adaptation and
loss and damage relevant time scales, i.e., up to 2050. It is important to be cognisant of the fact that on adaptation relevant
timescales until mid century, climate uncertainties dominate over scenario uncertainty (Lehner et al., 2020; Lee et al., 2021).
So even a very high emission pathway would not allow for a full appraisal of high-end outcomes on those timescales. Rather,
an assessment of high-risk outcomes needs to be based on assessing higher risk percentiles of any given pathway. In principle,
there are two options to examine the high-end warming futures for these purposes: Either to use a high end pathway, such as
'the emission world avoided' scenario proposed here like SSP5-8.5 or SSP3-7.0, and then - by examining impacts at different
global warming levels (GWLs) - mapping climate characteristics onto the high percentiles of expected tails of the distribution





of lower emission pathways, like the 'current policy' pathways. Or to use the higher-end warming tails of the ESM models that were run for those 'current policy' pathways, acknowledging that a lack of high ensemble sizes and a lack of a complete representation of uncertainties might hinder a full examination of the tails of the projected distribution. In contrast to the high

end of the scenario space, the plausible and very low end of the scenario space is a prerequisite to derive minimal adaptation needs and identification of unavoidable loss and damage. Understanding the very low end of the scenario space is also key to understanding the consequences of delayed action i.e., what we are leaving behind or taking off the table. Thus, a useful framing pathway is one in which immediate action had started in 2015 and that has at least a 50% probability of decadal average temperatures remaining below 1.5°C.


### 2.3  Non-exhaustive list of policy-relevant questions.

We postulate a non-exhaustive list of policy-relevant questions. The degree to which the framing pathway design can address those questions or not will influence to what degree the framing climate pathway dataset can help the forthcoming seventh assessment cycle of IPCC (AR7) and the wider scientific literature based on CMIP7 to be policy relevant.


a.  **What are the potential climate outcomes of current climate policy targets?** The climate ambition reflected in the current set of National Determined Contributions (NDCs) is insufficient to meet the Paris Agreement temperature goal (Riahi et al., 2022; Meinshausen et al., 2022). However, beyond the near-term, different interpretations of the stringency and credibility of expressed net zero targets provide for a broad range of different emission trajectories and subsequent

warming outcomes ranging from projected warming between 2.5°C and 3°C by 2100 (and continuation thereafter) to a pathway in which median peak warming is limited to less than 2°C (Rogelj et al., 2023). Exploring this range is key to inform the implementation and refinement of established targets.

b.  **What mitigation is required to limit warming to around 1.5°C and what climate impacts can still be avoided?** The design criteria would be strong mitigation in line with the Paris Agreement and national targets (such as net zero $CO_2$ by

2050). Only by exploring such a lower emission future in the ESMs, an integrated and informed decision making can be supported to pursue synergies of mitigation and adaptation actions in line with the Paris Agreement. It might be that such a pathway with net zero $CO_2$ emissions around the middle of the century, limited cumulative emissions until then, and net zero GHG emissions in the second half of the century (as Art. 4.1 of the Paris Agreement aims to), is at the very low end of future emissions that some integrated assessment models can produce[5]. ESMs themselves would also help inform

confidence in the lower warming bound, providing a process-driven representation of the climate outcome of a maximally strong decarbonisation and land-use strategy proposed in the scenario. Such low-end scenarios are paramount to estimate

---

[5] Such a low end is here assumed to loosely be defined by the maximal speed of the transition towards a net-zero future (and beyond) from a technological and resourcing point of view - which will differ in each model's implementation of the latest cost, technology and deployment information. This lower bound is subject to uncertainties and definitional choices.





what the remaining carbon budget is and what climate impacts might still be avoidable and to inform questions of climate justice.

c. **Within the range implied by the Paris Agreement temperature goal, what are the different feasible mitigation strategies?** The extent to which a temporary overshoot above global warming of 1.5°C, always constrained by holding warming to 'well below' 2°C, is regarded to be in compliance with the Paris Agreement differs among different policy stakeholders (e.g., Mace, 2016). While some interpretations have been suggested (Schleussner et al., 2022) the exact definition of 'well below' 2°C is also not established yet in the policy domain. Not pre-empting those decisions requires a set of scenarios with different median peak warming broadly within the range of temperature levels referred to in Art. 2.1 of the Paris Agreement (i.e., between 1.5°C or below and less than 2°C). This might include distinguishing peak and decline pathways from stabilisation pathways at 1.5°C, or at levels above 1.5°C but less than 2°C.

d. **What are the interlinkages of Paris Agreement compatible climate action and a broader sustainability agenda?** Climate and environmental policy aims for a broader set of sustainability objectives beyond emission outcomes alone. Many of those interlinkages with sustainable development goals (SDGs) are explored in the framework of different SSPs, but insofar they relate to different land-use futures will also be of direct relevance for climate outcomes. Global sustainable land futures will be decisively different from a continuation of unequal trends, both in terms of greenhouse gas emission trajectories, but also for land-cover change, water and fertiliser use, biodiversity etc. (Humpenöder et al., 2022). Exploration of such sustainable scenarios is of direct relevance not just for informing climate policy, but also other fora such as the discourse on biodiversity.

e. **What are the consequences of delaying mitigation?** In the Glasgow Climate Pact, countries recognized the need for accelerated action in this critical decade (UNFCCC, 2021). Yet, a yawning gap remains between 2030 emission levels implied by current Nationally Determined Contributions under the Paris Agreement (as of 2023) and 1.5°C pathways (Shukla et al., 2022). To inform the ambition expressed in the Glasgow Climate Pact, robust scientific information on the consequences of delaying stringent climate action is of key policy relevance. Without an immediate and a delayed scenario that aim to reach, for example, 1.5°C by the end of the century, policy makers will not be able to obtain that information on the consequences of delay from dedicated impact studies that are dependent on ESM runs. These ESM runs will be fundamentally important to enable, for example, the impacts and adaptation limits assessment of IPCC AR7.

f. **Non-$CO_2$: What are the effects of non-$CO_2$ mitigation?** Addressing non-$CO_2$ GHGs, in particular methane, has risen in prominence in policy circles. For example, the Global Methane Pledge, supported by more than 150 countries, specifically focussed on stringent methane emission reductions by 2030. Distinguishing between different emission strategies with focus on shorter or longer-lived greenhouse gases, and/or air pollution (namely aerosols, ozone and their precursors) reduction is of policy relevance. During CMIP5, RCP-driven runs, non-$CO_2$ emissions, and in particular $SO_2$ emissions, were rather closely aligned across the low to high end scenarios. The SSPs somewhat corrected for that with consideration of various air pollution controls derived from the overarching SSP narrative and show a wider variation across both non-$CO_2$ GHGs (and their precursors) and aerosols (see Figure 2 in Cross-Chapter Box 1.4 and Figure 6.18





in IPCC AR6 WG1, Masson-Delmotte et al., 2021; see also Gidden et al., 2019a). However, the sustainable pathways included both strong climate change mitigation and air pollution control preventing disentanglement of the co-benefits of climate change mitigation policies from actions focussed on air pollution (Szopa et al., 2021). It would be useful to investigate the implications of different gas-to-gas emission strategies, including regionally varying mitigation policies to
capture geographical patterns of short-lived climate forcer emissions, both in the framework scenarios as well as in dedicated sensitivity scenarios in additional MIPs (Persad et al., 2022; Persad et al., 2023). Furthermore, the potential for non-$CO_2$ greenhouse gases to be emissions-driven is emerging in some ESMs (Folberth et al., 2022).

g.    **Land based carbon dioxide removal: What are the relative risks and the effectiveness of different negative emissions strategies?** RCP-based designs do not allow to assess significant uncertainties around land based carbon
dioxide removal, because emissions fluxes are calculated in the context of Integrated Assessment Models using semi-empirical land carbon accounting that does not represent climate impacts on the strength of the land and ocean carbon sink and on nature-based solutions overall (Pörtner et al., 2022). Complex carbon process representations used in land surface components of ESMs are increasingly able to capture dynamics associated with carbon fertilisation effects, plant demographic responses, heat stress, drought response and fire risk which could potentially radically alter the capacity for
land-based negative emission fluxes. Defining scenarios in terms of fossil emissions and land use patterns or decisions allows these processes to be represented directly.

h.    **Overshoot: To what extent are the impacts of climate change reversible and on what timescales?** Significant differences in mitigation requirements in the second half of the century between different classes of overshoot scenarios have been identified in the latest IPCC assessment (as e.g., reflected in the IPCC WG3's C1 and C2 category, (Riahi et al.,
2022)). Information on the climate system consequences and related impacts of overshoot, however, is not comprehensively available. Exploring the differences between different policy relevant overshoot scenarios with ESMs is critical to address this gap.

i.    **Providing a backdrop of 'the world that could have been'.** In order to calculate benefits of past and future climate action and inaction, it is useful to have hypothetical backdrop scenarios, both at the high and the low end. The 'world that
could have been' at the high end could be a medium-high emission scenario, such as SSP3-7.0 from the CMIP6 cycle - depicting a future that unfolds in the absence of climate policies and in the absence of economic shifts that were largely kicked off by initial climate policies (like 'learning-by-doing' renewable technology cost reductions). At the low end, a 'world that could have been' can provide a hypothetical backdrop of what would have been still possible, if countries had acted fast enough to strictly stay below 1.5°C warming at all times starting in 2015 (when the 1.5°C goal was first adopted
within the UNFCCC context in the Paris Agreement). Such scenarios will be vital to inform an emerging policy discussion around loss and damage under the Paris Agreement, as it provides a reference point of what loss and damage could have still been avoided.

j.    **What could a worst-case outcome world look like in 2100?** Due to different types of uncertainties (biogeochemical, but also societal and geopolitical), it is important also to understand a low-likelihood but high-warming outcome (Lee et





al., 2021). A more elaborate way to explore the tails of the warming distributions of the scenarios seems warranted (Kemp et al., 2022).

k. **What are the climate effects of different regional emissions?** For shorter-lived air pollutants, in particular aerosols, as well as for land-use changes, regional climate effects can markedly differ for geographically variable forcings - and emissions from various regions will have different regional and global implications (Seneviratne et al., 2018b; Persad et

al., 2022). Also, some short-lived forcers have physically distinct mechanisms of interaction with the global climate, such as directly absorbing sunlight or altering clouds, perturbing precipitation and large scale circulation and have distinct effects from well-mixed greenhouse gases (Tang et al., 2018; Liu et al., 2018; Sillmann et al., 2019; Persad et al., 2023).

l. **What is the time-dependence of climate change impacts?** How does the climate evolve even under near-stable global temperatures? Net-zero $CO_2$ emissions are expected to result in slow-changing global temperatures (King et al., 2021;

MacDougall et al., 2020) but understanding of regional climate and long-term bio-geophysical impacts of net-zero $CO_2$ emissions remains limited. There is currently a large dependence in the literature of global warming level-based projections on sampling from fast-warming scenarios such as SSP5-8.5 (e.g., AR6 Interactive Atlas). Exploring how regional climates and different aspects of global climate evolve at a stable Paris Agreement-aligned global warming level (e.g., Sigmond et al., 2020; Mengel et al., 2018; Seneviratne et al., 2018a) would improve understanding of the effects of

climate stabilisation and inform policy and decision-making.

m. **What are long-term implications beyond 2100?** Scenarios supplied by integrated assessment models have focused on a time horizon of 2100, which has been consistent throughout the evolution of scenarios from SRES to RCPs to SSPs. In the early 2000s this was considered appropriate, but 2100 is now less than one human lifespan into the future and the life cycle of a lot of new infrastructure (Lyon et al., 2022). Several Earth system consequencesincluding sea-level rise (Mengel

et al., 2018), ice sheet loss and carbon cycle dynamics, as well as impacts to the natural system will continue beyond 2100, even in strong mitigation scenarios. While RCPs and SSPs contained extended scenarios for running ESMs beyond 2100 (Meinshausen et al., 2011; Meinshausen et al., 2020), they were not based upon detailed scenario modelling provided by the IAM community and, particularly in CMIP6, had low take up amongst modelling groups (Lee et al., 2021). Irreversibility is also coupled with the question of overshoot within this century and beyond (Frölicher and Joos, 2010).

## 3    Science questions for framing climate pathways

In this section we focus on key scientific questions for the design of future framing pathways. A much broader reflection on a vision for the future of climate modelling is, for example, expressed in a recent WCRP 2022 workshop and its meeting report (WCRP, 2023). This vision explicitly identifies a priority for co-design approaches with users and key partners, including the IPCC and other assessment communities, and a focus on deep mitigation scenarios for ESM scenarios.






A number of key scientific frontiers can be identified that, in some cases, overlap with the policy questions identified above. From our perspective these include, but are not limited to the ones listed below. Many of the scientific advances can be expected from specific sensitivity pathways that are more of hypothetical nature, such as excluding or including a certain forcing agent, pulse response, abrupt change and other idealised experiments. Some of these aspects are however also relevant for the design

of framing pathways. Those scientific research questions are – inter alia:

    a.  **What is the timescale of emergence of mitigation benefits?** While questions of overshoot and zero emissions commitment relate to long-term climate outcomes, the question of the emergence of mitigation benefits in the near-term is of importance to adaptation and loss and damage policy. Outlining and understanding, when, how, and which benefits of mitigation emerge is the basis to inform what impacts of

climate change can still be avoided (Ciavarella et al., 2017; McKenna et al., 2021; Samset et al., 2020). This requires a specific focus also on mitigation of non-$CO_2$ GHGs (Lanson et al., 2022; Samset et al., 2020),the regional climate effects of aerosols (Persad et al., 2022), disentangling the effects of air pollution policies from GHG reduction policies and a more clearly-defined non-mitigation counterfactual scenario.

    b.  **What is the zero emissions commitment (ZEC)?** One of the most central science-based benchmarks for

climate policy is the focus on achieving net zero $CO_2$ emission targets to halt global warming. While no further warming for net zero $CO_2$ emissions is a robust central estimate identified in the AR6, the uncertainties around that central estimate remain substantial, as well as how the zero emissions commitment might change as a function of cumulative emissions at the point of net zero $CO_2$. A very robust understanding of this zero emissions commitment (Jones et al., 2019; MacDougall et al., 2020) and how it relates to realistic

net zero transitions should thus be a central objective of CMIP7. Likewise, a robust understanding of the implications of achieving and sustaining net zero GHG emissions (as part of Art. 4.1 of the Paris Agreement) under the GWP-100 metric would be a useful and very policy-relevant insight (Schleussner et al., 2022).

    c.  **Climate system and carbon cycle feedbacks under overshoot.** Whether or not global temperature increase is indeed reversible strongly depends on the response of the climate system and the carbon cycle (Schwinger

and Tjiputra, 2018; Melnikova et al., 2021). Some feedbacks, such as permafrost melt, will continue over centuries even under a return of warming following overshoot (Gasser et al., 2018). More generally, the response of natural emissions is highly uncertain and can also drive the evolution of several non-$CO_2$ climate forcers with consequences for biogeochemical climate feedbacks and health, threatening to counterbalance the air pollution control efforts in some places (Szopa et al., 2021). At the same time, feedbacks of carbon

dioxide removal (CDR) methods on the carbon cycle need to be taken into account (Melnikova et al., 2021). A systematic exploration of those feedbacks is required to critically assess the potential and risks under overshoot scenarios.

    d.  **What changes in the climate system are reversible and which are not?** An emerging body of science indicates that impacts of climate change will continue beyond halting global warming or even overshoot.





This is most likely for time lagged systems such as sea level rise (Mengel et al., 2018), but also potentially for circulation patterns, rainfall and climate extremes (Pfleiderer et al., 2023). The question if and under what conditions irreversible thresholds of ice sheets or other systems may be crossed is also key (Wunderling et al., 2023). This is increasingly not a question of comparing very high to low warming scenarios, but emerging evidence suggests significant increases in risks between 1.5°C and 2°C that require this range to
be resolved very thoroughly.

e.   **How do consequences of land-based CDR compare to potential impacts avoided under overshoot scenarios?** The impact of wide-scale land-based CDR (e.g., afforestation, biomass production, enhanced weathering) for land surfaces will reach beyond impacts on biodiversity and food security, and can also contribute to changing albedo and non-$CO_2$ emissions (Fuss et al., 2018) or regional weather patterns
(Pfleiderer et al., 2023). Global warming and related impacts on terrestrial ecosystems and their uses, particularly on their disturbances such as fires, drought, and pests (Westerling et al., 2006; Liu et al., 2023; Canadell et al., 2021) can also influence the durability of these CDR interventions in ways that are poorly represented in IAMs today. Investigating futures with ESMs in which more or less emphasis is placed on offsetting residual emissions with CDR can provide vital insights as to the implications of some mitigation
or removal strategies.

f.   **Fidelity of ESMs against observed climate change.** Understanding emergent constraints had a central role in IPCC AR6 to constrain future projections (Brunner et al., 2020; Tokarska et al., 2020; Ribes et al., 2021; Liang et al., 2020). The historical realism of CMIP7 ESM runs will be key and likely an iterative process when scanning different forcing agents and their (combined) effects. To what degree are performance
metrics, i.e. the agreement of ESM output with observations, useful to learn about global and regional futures?

g.   **Change of ESMs from generation to generation.** A clear comparison point to measure advances and difference in ESMs from CMIP5 to CMIP6 was missing, as the underlying scenarios differed substantially. This proposal could address the need for having at least one overlapping scenario by using a previous
generation scenario as the 'the emission world avoided' (TEWA) scenario, such as SSP3-7.0 or SSP5-8.5 for example. This strategy would also allow for extended time for climate impact assessment, which has been historically difficult to achieve in the timeline between the delivery of ESM scenarios and the IPCC assessments (WG II in particular, as noted in Pirani et al., (submitted)).






## 4 A perspective on the next generation of framing climate pathways for ESM runs

In the following, we show and describe – as one contribution to the community discussion – key characteristics and categories of framing pathway that could inform the selection of framing pathways to be covered by the Earth System Model framing pathways under the CMIP7 initiative that could address the policy-relevant and research-oriented questions discussed above.

Key characteristics are:

    a. Explore the full range of long-term outcomes of these pathways under overshoot and time-lagged feedbacks this century and also beyond. A default extension of scenarios to 2150 may be pragmatic, while providing meaningful, and potentially diverging extensions on even longer timescales (until 2500) to explore a range of different very long-term futures.

b. Pathways should be emissions-driven and land use strategy-driven for $CO_2$. Concentration-driven pathways can still be included in the wider CMIP effort to allow for a more systematic intercomparison of emission and concentration driven approaches. Non-$CO_2$ GHGs, especially $CH_4$ and $N_2O$, would still likely concentration-driven for the bulk of ESMs - given the nascent field of interactive gas cycles in ESMs (Sanderson et al., in prep.).

We suggest categories that should be represented by a representative emission pathway, starting from the highest emission category (Table 1):

- **TEWA**: **'The Emission World Avoided'**, a category for a pathway with high or very high emissions

- **NFA: "No Further Action"**, a category for a pathway reflecting current emission futures in the absence of any further climate action. The pathways in this category should ideally be accompanied by a perturbed physics ensemble
(acknowledging that those come with additional challenges of drifts and/or flux corrections etc., e.g., Shiogama et al. (2012)) as this would allow us to obtain valuable proxies for a worst-case high-end warming outcome under emissions implied by current policies (low likelihood / high impact).

- **DASMT**: **'Delayed climate Action and Stabilisation pathway Missing Target'**, a category for a pathway that misses the Paris Agreement long-term temperature goal as it results in global warming of around 2ºC in 2100, rather
than staying "well-below" 2ºC. Such a pathway explores global emissions approximately in line with NDCs and long-term targets as they were proposed around the time of Glasgow, COP26, (approximately resulting in a median just below 2ºC warming, if fully implemented)[6].

- **DAPD**: **'Delayed Action Peak and Decline'**, a category for a pathway in which climate action is further delayed, but that then features rapid emission declines and strongly negative long-term $CO_2$ emissions.

- **IAPD**: **'Immediate Action Peak and Decline'**, a category for a pathway that features immediate 2025 onset of decisive emission reductions and achieves net zero $CO_2$ emissions by mid-century.

---

[6] assuming an AR6-like assessment of climate characteristics (Meinshausen et al., 2022; Rogelj et al., 2023).

**IA2015: 'Immediate Action in 2015'**, a category for a pathway that resembles 'a world that could have been' at the low emissions end, assuming emission reductions towards net-zero had started in 2015. Other 'world that could have been' scenarios can be envisioned, and could be policy relevant, e.g., one starting in 1992 with the establishment of

the UNFCCC, or 2009.

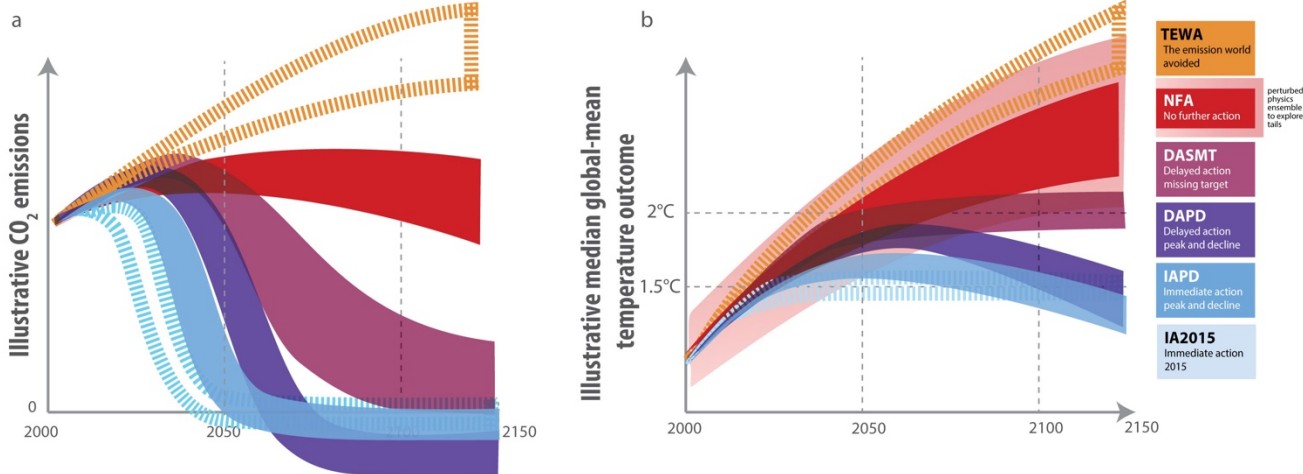

**Figure 2 - Sketch overview of framing pathway categories for ESM runs.** Illustrative $CO_2$ emission trajectory ranges and corresponding global-mean temperature outcomes are shown in panel a and b, respectively. Shown temperature outcomes are approximate median outcomes under IPCC AR6 assessed climate characteristics. In predominantly emission-driven runs, the range of temperature outcomes would vary
even further across models, because pathways are defined by their carbon emissions in conjunction to forcing series for non-$CO_2$ driving forcers and because different models exhibit different responses. Categories that represent hypothetical edge pathways ("the worlds that could have been") are possibly of lower priority (illustrated by dashed lines), but frame the pathway space at both the higher emission end and lower emission end. The NFA category pathway should ideally also be run by perturbed physics ensembles and other approaches to capture the full uncertainty and its tails of warming so that a higher warming outcome representation of that REP could serve as a proxy for
a high-impact, low-likelihood scenarios ('see background red range' for NFA category). This would be particularly important for adaptation and risk assessment. If a perturbed physics ensemble high-end warming outcome of the NFA scenario is not able to be investigated, the high 'the emission world avoided' TEWA category could also provide a proxy for a high-end / worst-case warming outcome under lower emission futures.






**Table 1 - Overview of suggested pathway categories to inform the design of specific representative emission pathways for Earth system model runs. Categories of emission pathways identified in the IPCC AR6 WGIII (compare Table SPM.1) and selected WGI core SSP-RCP scenarios are provided for comparison. For each category, we provide an indicative 'priority' suggestion, recognizing**
**that there are limited resources to run a large set of scenarios across all ESMs.**

| Category to be represented | Key characteristics of the representative pathway | Advantages | Potential drawbacks | Closest category (and selected pathways) in IPCC AR6[a] |
|---|---|---|---|---|
| **High emissions: "TEWA" - The emission world avoided** | • High end emissions<br>• Departing from historical emissions in the past, i.e., 2015.<br>• Three main options are SSP5-8.5, SSP3-7.0 or a new pathway that retrospectively reflects 'no-further climate action' (NFA) starting in, e.g., 1992, 2010 or 2015, each with their respective advantages and challenges (aerosols, comparability to CMIP6 and possibly CMIP5, representativeness of previous reference scenarios, etc.)<br>• Lower priority | • Allows depiction of the world that could have unfolded without climate policies.<br>• Allows to learn about high tail warming possibilities of lower scenarios.<br>• Allows direct comparison of new generation of ESMs with previous ones, if a CMIP6 high end pathway is repeated (SSP5-8.5 or SSP3-7.0).<br>• High signal-to-noise for projected changes in climate | • Could be mistaken as a reference case pathway.<br>• Could lead to the false impression that the difference with this avoided scenario is the exclusive result of successful climate policies, and therefore that we have already achieved the biggest part of the challenge and what is left requires a smaller effort in comparison. | AR6 WGIII category C7-C8<br><br>SSP3-7.0 or SSP5-8.5 or RCP8.5. |
| **"Medium" or "No further action (NFA)"** | • A medium-high category that approximately reflects the median of "current policies as of 2023" or "current trends" estimates.<br>• Approximately flat global GHG emissions from 2025 towards the end of century.<br>• Approximately resulting in a 2.5-3.0°C world by the end of the | • An approximate depiction of future emissions in the absence of further climate policy action and assuming continuation of "current trends" as of the early 2020s.<br>• Reflective of 2°C crossing up to approximately 2.5-3°C warming by 2100. | • The longer-term evolution of emissions under current policies is highly uncertain. Together with the DASMT this category spans the range of future policy outcomes (as of 2023) | AR6 WGIII category C6<br><br>SSP2-4.5, RCP4.5. |





| | | | |
|---|---|---|---|
| | 2100 under median ECS/TCR.<br>• Encouragement to ESMs to provide a perturbed physics ensemble to obtain a high-impact, low-likelihood warming future relevant for risk assessments.<br>• A further extension of a pathway from this category beyond 2150 would be helpful for investigating tipping elements or slow response in the Earth system such as ice sheet or permafrost.<br>• A higher priority pathway | • Might allow to depict progress of climate policies over the last decade when compared to the 'TEWA' scenario,<br>• Key for climate risk assessment to inform adaptation with or without considering high-end tail risks.<br>• A high-end warming outcome for a NFA pathway could be framed as a high-impact low-likelihood pathway. | | |
| **"Delayed action and stabilisation, but missing target" (DASMT),** | • A category reflecting the most optimistic end of current climate targets<br>• A higher priority pathway | • Approximately reflects full implementation of all the emission targets currently proposed with expected median levels around 2°C by the middle and end of the century.<br>• Could be reflective of net-zero $CO_2$ by the end of the century. | • The NDC and long-term targets are going to be enhanced, which might render this DAP pathway category not fully aligned with targets "as of publishing date of AR7" | AR6 WGIII category C3b<br><br>SSP1-2.6 or RCP2.6<br><br>IMP scenario 'IMP-GS' (Shukla et al., 2022) |
| **"Delayed action peak and decline (DAPD)"** | • A category to be represented by a pathway with the same near term (up to 2030 or 2035) emission trajectory as DASMT.<br>• Pursues net-zero $CO_2$ emissions by the middle of the century (2050-2060) and net-zero GHG emissions around 2070-2080. | • A high overshoot and strong net negative $CO_2$ emission world.<br><br>• Providing a comparison scenario to the low overshoot 1.5°C scenario with similar full-century cumulative $CO_2$ emissions. | • Very strong net negative $CO_2$ emissions imply potentials for strong changes in land use patterns.<br>• Different pathways in this category might lead to different climate outcomes due to different land | AR6 WGIII category C2<br><br>IMP scenario 'IMP-NEG' (Shukla et al., 2022) |



| | | | | |
|---|---|---|---|---|
| | • Strongly net-negative GHG emissions thereafter<br>• A high-priority pathway | | use patterns depending on whether BECCS or DACCS or other CDR options are emphasised | |
| **"Immediate action peak and decline (IAPD)"** | • A category to be represented by a pathway with strong global emission reductions from 2025 onwards towards global net-zero $CO_2$ by 2050<br>• Strong emphasis on non-$CO_2$ GHG emission reductions, in particular methane.<br>• Implementation of a broad sustainable land-use agenda including demand side measures and dietary changes<br>• Approximately a low overshoot 1.5°C scenario.<br>• A higher priority pathway | • Could be representative of the 'best possible' mitigation future and 'adaptation minimum'.<br>• Strong methane reductions could reflect enhanced 'global methane pledge' ambitions.<br>• Sustainable land-use focus allows for policy relevant insights when compared with DAPD<br>• Similar to SSP1-1.9, allowing for comparison at the lower end of the scenario spectrum with CMIP6. | • Potentially climatically close to the DAPD pathway, but the strong methane reduction and land-use differences could set this scenario clearly apart | AR6 WGIII category C1<br><br>SSP1-1.9, IMP scenario 'IMP-SP'(Shukla et al., 2022) |
| **"Immediate action starting in 2015" (IA2015)** | • A lower priority pathway category<br>• The world that could have been if Parties had commenced immediate global action from 2015 onwards towards net-zero $CO_2$.<br>• Potentially building on SSP1-1.9 | • Approximately reflective of a 1.5°C pathway without overshoot.<br>• Can serve as a baseline to the low and high overshoot scenarios and inform assessments of loss and damage<br>• Serves as reference 'stabilisation' pathway to compare with DASMT, with the latter stabilising at approximately 0.5°C higher warming around 2°C. | • Potentially climatically close to the IAPD pathway | SSP1-1.9 |

[a] No socio-economic storylines should be associated with the next generation of pathways, similar to RCPs.





Focusing on largely emission-driven runs while at the same time providing a framework that envisages certain global warming futures being investigated requires methodological clarification as ESMs will produce different future global surface
temperature outcomes for the same emission future. To a lesser degree that same issue existed in the previous generations of RCP and SSP climate pathways, as ESMs did not share the same internal radiative forcing for the same input dataset of concentrations, emissions and land use patterns, yet were all labelled 1.9W/m$^2$ or 2.6W/m$^2$ pathways. We propose two ways to address this issue: Firstly, the headline names of the scenarios do not include the temperature level, but rather a qualitative label on the emission pathway. Secondly, we suggest continuing the practice of using the previous IPCC assessment cycle's
findings (here, AR6) to design the scenarios for the next assessment cycle. For example, the SSP scenarios for the CMIP6 and the AR6 report, including the Special Report on Global Warming on 1.5°C, were selected to match 1.9W/m$^2$ or 2.6W/m$^2$ labels by using a default AR5-calibrated MAGICC version, even though AR6 was expected to advance our knowledge on carbon cycle, other gas cycles and radiative efficiencies. Thus, an SSP1-2.6 pathway under the AR6 assessed science on gas cycles and radiative forcing will neither necessarily result in a 2.6W/m$^2$ median forcing – nor will this forcing level be uniform across
the ESMs. Similarly, but somewhat more pronounced, we suggest that the emission pathways are designed using the AR6-calibrated climate emulators to match the design criteria – while at the same time acknowledging that the advancement of science will result in some shift of the best-estimate temperature projections (compared to what AR6-calibrated emulators produce).

## 5   Discussion

### 5.1   Framing pathway storylines in line with recent IEA and NGFS scenarios.

We note that the set of categories identified here – which are to be represented by specific framing pathways - bears some similarities to scenarios that have been identified by other, initiatives not necessarily related to the IPCC assessment, that are of key policy relevance, such as the scenario set investigated by the International Energy Agency (IEA) (IEA, 2022), or the Network for Greening the Financial Sector (NGFS) (Richters et al., 2022). In particular, those key pathways sets examine the
worlds under current policies, current climate targets on the higher side and emission levels in line with the ultimate goal of the Paris Agreement on the lower side. To us, this illustrates a convergence in views of what are policy relevant perspectives on pathways that the climate science community may want to take on board. The initial June 2023 workshop discussions for the forthcoming ScenarioMIP protocol, at which this paper's proposal was presented, also picks up some elements of this convergence of views (Figure 1 in van Vuuren et al., 2023). We hpe that the science and policy objectives, as well as the
presented pathway categories and timing considerations, outlined in this contribution can further inform the deliberations under ScenarioMIP, as well as other MIPs, under the CMIP umbrella.





## 5.2 High granularity of lower pathways categories.

We have identified three climate pathway categories between 1.5°C and below 2°C. One of the pathway categories (IA2015), a lower priority one, represents a low "the world that could have been" pathway, i.e. investigating a hypothetical world in 620 which global emissions would have diverted from historical emissions in 2015 to stay below 1.5°C. The other two categories, IAPD and DAPD (see Table 1 above), resemble the lowest two AR6 WGIII categories C1 and C2 that in the central outcome are expected to return global-mean temperatures to below 1.5°C by the end of the century again. One argument against a high granularity of ESM runs is often that they are too close to each other to detect climate differences. This presumption is in direct contrast with a high level message from the IPCC AR6 that "every bit of warming matters" and requires further reflection. 625 Firstly, differences between small increments of warming can indeed be detected (e.g., Pfleiderer et al., 2018) including for long-term sea level rise where differences in 0.2°C peak warming may amount to 40cm difference in 2300 sea level rise commitment or more (Mengel et al., 2018), and also in the near-term emergence of climate extremes on the country level when considering large ensembles (Beusch et al., 2022a). While for a single ESM framework, large ensembles are required to statistically investigate differences in "close by" scenarios or weak single-forcing differences (Shiogama et al., 2023; Smith et 630 al., 2022), the multi-model nature of CMIP could allow to investigate the multi-model average differences across close-by scenarios, without the need for each modelling centre to entertain very large ensembles. Also, while pathways might differ by not more than 0.2°C in terms of global-mean temperature, differences in regional emissions across those scenarios imply stronger differences in regional climate outcomes (e.g. Persad et al., 2023). In addition, the category characteristics outlined above would imply very different land-use futures. These differences would translate to different climate futures in particular 635 on the regional level and, given the differences in the representation of land-climate feedbacks across ESMs, also a range of different global climate outcomes that is of key importance to explore to inform the policy discourse on land-based mitigation.

In addition to differences in peak warming, very different overshoot outcomes would be implied under a IAPD in comparison to a DAPD – type future. A systematic exploration of those differences, and the robustness of different mitigation strategies 640 when considering climate impacts, is of critical importance given the profound differences between such pathways in the mitigation space. Whether we advance mitigation efforts to achieve a low overshoot instead of a higher one is a question of highest societal relevance. To meet the high policy interest around future temperature outcomes (that can be argued to be) within the long-term Paris Agreement goals, the AR6 WGIII report (Shukla et al., 2022) for example placed a strong emphasis on so-called illustrative mitigation pathway (IMP) scenarios that are largely in the lower 1.5°C with no or low overshoot 645 category. Similarly, we argue it is time to enable WG II to provide the corresponding impact assessment via adequate ESM runs that provide the needed geophysical input for the impact studies.

Lastly, sometimes the argument is made that emerging pattern scaling or regional emulator approaches could fill in and extrapolate ESM results. While that might be increasingly possible in the future (especially if proposed ESM experiments





provide good training datasets), the current set of regional emulators is however not yet able to fill that niche. That is particularly true for peak and decline (overshoot) scenarios that have the potential to exhibit hysteresis in terms of large scale and regional warming and precipitation patterns (Pfleiderer et al., 2023). For example, approaches that rely on global-mean temperatures as one of their input parameters, are not yet adequately capable of distinguishing warming, stable and cooling worlds, before, at or after a peak in global-mean temperatures (see point 5.6 below).

In summary, we suggest that future framing pathways separately explore pathways from an immediate action, sustainable future category (IAPD) and a delayed action, high overshoot, high negative emission category (DAPD). These could arguably also be seen as exploring different futures within the Paris Agreement temperature goal range. Having at least two scenarios will prevent a singular de-facto definition by the scientific community of what the Paris Agreement goal exactly means (which could be regarded as policy-prescriptive). Exploring pathway variants that yield to temperature pathways within the Paris

Agreement temperature goal range obviously belongs one of the most highly policy-relevant questions (Masson-Delmotte et al., 2023), and should be more than worthwhile the investment of resources.

### 5.3 High warming pathway

Our deliberations still include a high warming pathway category, i.e., 'the emission world avoided', - as those high warming pathways are widely used in the community and serve the scientific purpose of understanding climate change under large

forcings. In addition, the more idealised 1%CO2 and abrupt forcing runs adequately assist that scientific quest into better understanding Earth System characteristics in a high forcing / high warming world. A number of scientific applications, i.e., related to emulator calibration and global warming level assessments, would also continue to rely on such high forcing pathways and we do not argue that such high forcing outcomes should not be modelled in the next generation of ESM framing pathways. Our proposal however no longer includes a high warming pathway category that could be mistaken as a 'business-

as-usual' scenario and we argue that it would be beneficial to separate high forcing pathways for scientific purposes, from the more policy oriented framing pathway categories.

The high-end 'emission world avoided' pathway at the upper end could also serve a strong communication purpose. Frequently, the success of the Montreal Protocol in limiting the emissions of ozone depleting substances is showcased by comparing current emissions to "the world avoided" scenarios (Velders et al., 2007). Having a similar comparison point or range in climate

science would be a useful indicator of where we might have been if we had failed to put climate action on the political agenda, In contrast, at the other end, exploring the low-end emissions 'world that might have been' is also a reminder of what we could have achieved if not for political, and economic forces that inhibited swift global-scale emission reductions over the last decades (Supran et al., 2023).

### 5.4 Separate consideration of socio-economic pathways.

It is worth noting that this proposed set of scenarios does not prescribe specific socio-economic futures but can accommodate different narratives of socio-economic development as reflected in the original Shared Socio-Economic Pathway (SSP)





framework, as well as different perspectives on burden sharing, equity and fairness that have been identified as a key element of scenario development moving forward coming out of the Bangkok discussions. The one methodological challenge will be to derive characteristic land-use and aerosol precursor emission patterns that are representative of some future socio-economic evolutions, with potential variations then studied in additional sensitivity MIPs (Section 5.5).

### 5.5  Framing pathways to be complemented by sensitivity MIPs.

A limited set of climate 'framing pathways' should be complemented by a much broader set of explorations in different MIPs under the new round of CMIP. Those sensitivity cases should explore land-use patterns, different aerosol assumptions (both spatially and different time series evolutions), methane reductions, global warming levels etc. But in our view the climate 'framing pathways' presented here would provide for a good basis for interlinkages with different MIP explorations on those key topics and key research questions as outlined above. We also note other, potentially policy relevant emerging approaches for scenario design beyond emission or concentration driven runs that explore adaptive, ESM specific approaches to match different warming level outcomes, thus filling a critical gap in explicitly exploring warming level dependent impacts in ESM space (Terhaar et al., 2022).

### 5.6  Regionally explicit emulators are still in early stages.

Separate from regional downscaling exercise that benefit from a long tradition and use ESM output as a starting point, the question has emerged to what degree regionally explicit emulators can replace ESM runs. As aforementioned in Sec. 5.2, the use of emerging tools like spatially explicit emulators in combination with well-established reduced-complexity climate models (e.g. Beusch et al., 2020; Beusch et al., 2022b; Tebaldi et al., 2022) can provide for very promising applications to explore a range of different futures in rapid fashion. However, such approaches, often based on the idea of pattern scaling of global-mean temperature, can only be used for emulation of ESM outputs, and without adequate ESM training pathways will not be able to capture key features of peak-and-decline or stabilisation pathways. Furthermore, none of those emulators is yet able to provide the full richness of variables, spatial and temporal and cross-variable correlations that ESMs are able to provide - and which are key for a range of specific impact models like those linked together in the ISIMIP project (Frieler et al., 2023; Warszawski et al., 2014). Thus, while emulators will be able to play an increasingly important role, they do not provide sufficient capabilities to address the science and policy questions identified above. ESMs are still the key method of choice to synthesise, diagnose and analyse our best available climate system science under various driving forces. This might change at some point, but it would be a large risk to bet that emulators will be mature enough to fill our key scenario gaps by the time of AR7.

### 5.7  Linking projections of GHG and ozone-depleting substances (ODS), as well as new gases.

To date, projections of the effects of atmospheric accumulation of GHGs, ODSs, and ODS replacements (hydrofluorocarbons, HFCs) have occurred primarily through two communities, the IPCC for GHGs and the WMO/UNEP Ozone Assessments (e.g.





World Meteorological, 2022; Velders and Daniel, 2014) for ODSs and HFCs. While there has been increasing connectivity and collaboration between these two communities, projections of GHG and emissions of ODSs and their replacements have
often been generated through substantially independent frameworks resulting in the underlying emissions 'storylines' often not being consistent. For example, in a given ozone assessment, the baseline ODS scenario from the previous assessment has been commonly used as the baseline for the needed 3-D model calculations. This can lead to inconsistencies when IPCC assesses and uses a new set of ODS emissions scenarios to quantify the direct and indirect radiative forcings of ODSs and their replacements, with these scenarios often being distinctly different from any WMO/UNEP ODS scenarios. Furthermore, even
the framework scenarios assessed by the of IPCC scenarios, whether to attain an overall radiative forcing by some year or to follow a prescribed global development path, is not consistent with the approach taken by WMO/UNEP assessments. Given this and given that ESMs now increasingly include an interactive stratospheric chemistry scheme that explicitly couples the chemistry-climate effects of atmospheric GHG and ODS loading, a set of self-consistent GHG and ODS emissions scenarios would be another opportunity to link these two research communities and provide single set of simulations that can serve the
needs of both communities, as well as benefit their intended policymaker stakeholders. As industrial processes, energy generation, and transport activities transition to GHG and ODS alternatives, it is essential that the next generation of REP climate pathways includes the new comp emitted from the associated activities (e.g., the effects of fugitive hydrogen emissions on atmospheric chemistry, the stratosphere and climate from a hydrogen-based economy (e.g., Tromp et al., 2003)).

**5.8  Timely provision of updated pathway information to inform the global stocktake**

Countries striving to implement their climate plans, including mitigation and adaptation objectives as well as responses to loss and damage, will require significant investments in the envisioned economic and societal transformation (Shukla et al., 2022). To inform policy and investment decisions, timely provision of scientific analysis and assessments is key. In response to this demand, actors such as the IEA or the NGFS provide annual updates of emission pathways. Such a rigid timeline is of course out of reach for community pathway developments focussed on ESMs. However, we would argue that a reflection on the
timeline for pathway design in the light of stakeholder needs, and in particular political processes under the Paris Agreement, is very much in order.

The second global stocktake (GST) under the Paris Agreement will be conducted in 2027/2028 (Figure 3). Informed by its outcomes, parties to the Agreement will be invited to develop their NDCs for 2040. A 2040 horizon is close to expressed net-zero $CO_2$ and net-zero GHG targets for several major emitters, and very deep emission reductions, as well as building up
carbon dioxide removal capacities will be required to achieve them (Edenhofer et al., 2023). A number of the policy and science questions identified above are of direct relevance to inform the development of NDCs on this timeline. Striving to provide major scientific community inputs in time for the 2[nd] GST will be essential. Thus, an enhanced timeline of some CMIP7 outputs (or also potential continued CMIP6 activities to fill critical gaps in the existing scenario space) is worthwhile considering. It would be very unfortunate, if the best available science at the time of the second GST in 2027/28 would still
need to predominantly rely on a set of ESM scenarios dating back to 2015 (the current CMIP6 SSP-RCP framework).



Reflections on scientific inputs into the 2nd GST need to be cognisant of the timeline of the IPCC AR7, as the IPCC will be a key source of input to the Global Stocktake process. As the AR7 is about to start, we do not know with what products and in what ways the IPCC will inform the 2nd GST (but note discussions at the Bangkok meeting that suggested the need to consider producing relevant outputs by 2028 (Masson-Delmotte et al., 2023)). We certainly do not want to pre-empt any discussions on

that matter, but would argue that there is a good chance that the IPCC will aim for some kind of product to inform the 2nd GST. Figure 3 depicts a plausible range of when such a hypothetical IPCC product, or even a series of products, would need to be produced. From that range, we can work backwards with potential literature cut-off dates for any such a product. Based on this, we argue that aiming for providing ESM data by mid 2025, and therefore making forcing data available by mid 2024 would be required. This tight timeline highlights the need for early preparation and reflection by the scientific community well

ahead the finalisation of IPCC schedule. Furthermore, the challenges experienced in the AR6 with 'hot models' might caution against making simulations available too close to the deadline (Hausfather et al., 2022).

Missing the opportunity for input into the 2nd GST will not just forego the chance to inform one of the key processes for scientific input under the UNFCCC, but, and arguably even more importantly, also the time window where countries can still take on board new climate science insights in the preparation of their NDCs with a 2040 target year. Furthermore, some link

between the IPCC timelines and the international climate change deliberations between countries seems pertinent to fulfil IPCC's mandate to be policy-relevant – hence is necessary to maintain and build the IPCC's standing in the international discourse.

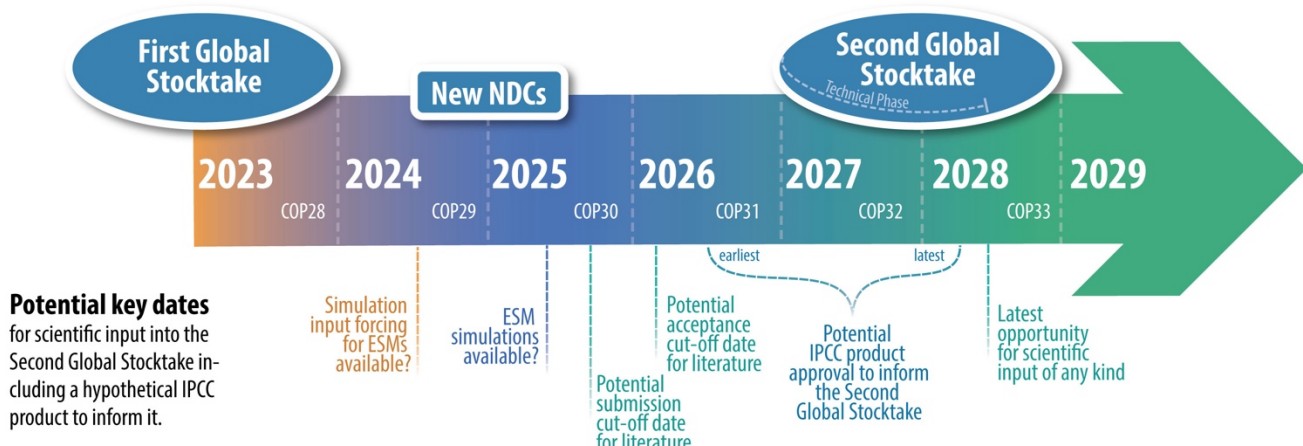

**Figure 3 – A hypothetical timeline for scientific information and IPCC to inform the 2nd Global Stocktake.** Key dates are derived

based on the global stocktake cycle and its modalities, including a technical phase for scientific input. It is hypothetically assumed that the AR7 will consider providing some kind of product to inform the 2nd GST. However, including such a hypothetical timeline for an IPCC product allows to establish a timeline for the design and execution of a new generation of Earth System Model simulations aimed to inform the relevant IPCC product. Note that cut-off dates are indicated relative to the earliest IPCC product approval timing and would move accordingly backwards for a later approval date.




## 6    Conclusion

A multitude of different factors inform scenario design processes for the next generation of ESMs. But in particular for the selection of framing pathways for coordinated ESM simulations, e.g., the SSP-RCPs in CMIP6, we suggest that policy considerations need to be fully taken into account given their relevance for the IPCC assessment, and to inform climate policy and action more generally. One key challenge is to not constrain the future climate space in a manner that excludes exploration of expressed policy objectives and hence might prevent forthcoming IPCC assessments from fulfilling their mandate to provide a policy relevant, yet not policy prescriptive assessment (Masson-Delmotte et al., 2023). Given ongoing scenario design process under the CMIP umbrella will effectively set the framing scenarios on which any forthcoming IPCC assessment will be based, considerations on how to avoid potential policy prescriptive choices by the research community  need to be considered very carefully, considering a comprehensive range of policy outcomes that include representation of the maximum ambition of the Paris Agreement (limiting warming to 1.5C) and potential high end outcomes..

When it comes to the representation of pathways that could be considered Paris Agreement compatible, it is important for science to be in the position to provide an open, transparent, and full appraisal of the co-benefits, efforts, and changes from past practices that might be warranted in order to pursue certain warming outcomes. A focus on the lower range of scenarios is also reflective of a post-Paris policy landscape in which questions are not anymore about whether or not climate targets will be set, but about the credibility, risk and implementation of proposed pathways to meet those targets as well as defining the necessary pace of climate action in order to achieve them (Meinshausen et al., 2022; Rogelj et al., 2023).  Informing such reflections also requires scientific information on the consequences of not achieving certain collective policy ambitions and process-based assessments of the implicit technological and physical assumptions in those pathways.

This progression in the climate discourse is informed also by one of the main messages from the IPCC's AR6 report series, that "every increment of global warming" matters. Further substantiating such an assessment will be of key importance for climate science and forthcoming IPCC assessments, which requires new framing pathways to provide sufficiently granular resolution, not just for end-of-century (or beyond) global warming outcomes, but even more so on the decision-making relevant horizon until mid-century.

As we argue in this contribution, science and policy questions for the next generation of framing pathways also include those related to a potential temperature overshoot and return, and the benefits, and negative side effects, as well as technological and physical plausibility of large-scale carbon dioxide removal. We further argue that due consideration should be given how a community effort for the development of new scenarios and related scientific insights can inform the 2nd Global Stocktake under the Paris Agreement and the development of new submissions of NDCs for a target year 2040.

Based on these considerations we have identified a range of categories that would, together, satisfy key considerations and could be represented by framing pathways called "Representative Emission Pathways" (REPs). Further considerations beyond those outlined here may inform the ongoing efforts of pathway design as part of CMIP, including ScenarioMIP (van Vuuren et al., 2023) and/or other model intercomparison projects (MIPs). We hope that our perspective can serve as an input into a



strategic approach for driving ESMs with policy-relevant futures in this critical and contribute to further community reflections on the next generation of framing pathways that we think will greatly benefit from an open and inclusive discussion given the far-reaching consequences for climate science and policy stemming from the scenario design.

## 7    Code availability

No code has been produced or used for this manuscript.

## 8    Data availability

No data has been generated and figures are illustrative only.

## 9    Executable research compendium (ERC)

Not applicable.

## 10    Sample availability

Not applicable.

## 11    Video supplement

Not available.

## 12    Supplement link

Not applicable.

## 13    Author contribution:

MM wrote the first draft and CFS edited the first draft based on discussions at the IPCC Expert Workshop in Bangkok, April 2023. Further development was co-led by MM and CFS. GPP and ZN provided a first sketch of the conceptual overview figure 1, implemented by MM and CFS. All authors contributed to the paper.



## 14 Competing interests:

The authors declare that they have no conflicts of interest.

## 15 Acknowledgements

The authors would like to thank Matthew Gidden, Alaa Al Khourdajie, Hideo Shiogama, Janna Sillmann, Anna Zehrung and other valued colleagues for very helpful comments on an earlier draft. CFS, BMS, AT, SS, JR, and JS acknowledge funding from the European Union's Horizon 2020 research and innovation programmes under grant agreement No 101003687

(PROVIDE). GPP acknowledges funding from the European Union's Horizon 2020 research and innovation programmes under grant agreement No 821003 (4C) and the European Union's Horizon Europe Research and Innovation Programme under grant agreement No 101056306 (IAM COMPACT). RS and ZN acknowledge funding from the European Union's Horizon 2020 research and innovation programmes under grant agreement No 101003536 (ESM2025). ADK, MM and TZ acknowledge funding from the Australian Government National Environmental Science Program.

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
