# Peer review of "A perspective on the next generation of Earth system model scenarios: towards representative emission pathways (REPs)"

_Geoscientific Model Development, 2023_

## Referee Comment (RC2)

**Meinshausen et al: A perspective on the next generation of Earth system model scenarios: towards representative emission pathways (REPs)**

**Review by Dale S. Rothman**

**Preface**

In preparing this review, I first went back and reviewed the SPMs from IPCC AR6, the report of the IPCC Workshop on the Use of Scenarios in the Sixth Assessment Report (Masson-Delmotte et al 2023), and the report of ScenarioMIP Workshop: Pathway to Next Generation Scenarios for CMIP7 (van Vuuren et al 2023). Furthermore, I considered the proposal for Representative Emission Pathways (REPs) in the broader context of scenario use in climate change analysis. Specifically, I have adopted the position that evaluating a proposal for a set of representative scenarios needs to keep in mind their potential use for various types of analysis. This points to the importance of being able to interpolate (not extrapolate), both spatially and temporally, the information provided by the representative scenarios in order to meet the needs of these other analyses. I have sub-divided my comments into three parts: 1) Issues with Representation of the RCP-SSP-SPA Framework, 2) Choice to use Emissions Pathways, and 3) Specific Emissions Pathways Recommended.

**Issues with Representation of the RCP-SSP-SPA Framework in AR6**

My comments here focus on a number of statements that I feel mis-represent the RCP-SSP-SPA framework. Specifically, the statements "the so-called SSP-RCP matrix (Moss et al., 2010; van Vuuren et al., 2014) was used to explicitly present the climate and socio-economic dimensions as independent dimensions" and "Shared Policy Assumptions (SPAs) were used to vary the climate outcomes (Kriegler et al., 2014)" on p. 6 concern me. While these issues are less important to the present paper as my later issues, they do need to be recognized.

The RCPs were developed early on as a quick way to provide quantitative emission/concentration pathways as inputs to ESMs and GCMs. The nomenclature, e.g., RCP8.5, was, unfortunately a bit confusing as 8.5 referred to the associated radiative forcing estimated in the IAMs rather than either the levels of emissions/concentrations. The pathways did have underlying socio-economic and land-use assumptions as inputs into the IAMs, which needed them to produce the emission/concentration pathways, but these were not intended to be examined in any great detail.

The SSPs were subsequently developed by thinking more carefully and consistently about the underlying socio-economic aspects, e.g., population, education, economic, and technological changes. This did help to produce some key quantitative results for each SSP using, among other things economic and education models, but also some more qualitative elements[1]. These were used as inputs to the IAMs, which were used to produce, among other things, new quantitative
* * *
[1] The qualitative elements were important for the SPAs as noted in the next paragraph, but also for further, usually IAV, studies that needed more details than provided by the quantitative information.

pathways of emissions, concentrations, and radiative forcing, also to be used as inputs to ESMs and (AO)GCMs. The initial SSPs excluded considerations of explicit climate policy, but later versions were developed to produce pathways with lower levels of radiative forcing. Although some studies depicted as using, e.g., SSP1-RCP1.9, did combine the socio-economic elements from an SSP with the emissions/concentrations/radiative forcing pathways from one of the original RCPs, the intention was that studies should actually use the emissions/concentrations/radiative forcing pathways produced from running the outputs of the IAM run with the assumptions from the SSP.

Finally, the SPAs were meant to reflect the fact that different climate policies made more sense in different SSPs and, therefore, make the SSPs including climate policies more internally consistent. For example, global participation did not really fit well with the story being told in SSP4. While this did lead to differences in the climate outcomes, it is inaccurate to say that this was their purpose.

**Choice to use Emissions Pathways**
Menhausen et al choose to focus on Representative Emission Pathways rather than pathways identified by either concentrations or levels of radiative forcing. They discuss some pros and cons of this choice already, but I would like to add a few additional comments.

On the pro side, using emission pathways would likely lead to 'cleaner' pathways coming out of the IAMs, which they note are "likely to be derived from IAMs". This assumes that the IAMs are able to provide these emission pathways prior to any changes resulting from their own internal carbon cycle and climate components and resulting feedbacks. Otherwise, there could be issues of conflicting effects with the carbon cycle and climate components and resulting feedbacks in the ESMs and (AO)GCMs.

On the con side, there are several issues, two of which I note here:

- Any single REP would likely be associated with any number of concentration, warming, hazard, and impact and risks pathways. The first three of these relate to the models used to estimate concentrations, warming, and hazards. As for the impact and risk pathways, these can differ given that it is likely that there could be a many-to-one relationship between socio-economic/policy pathways and emission pathways, as the former are important considerations in estimating impacts and risk.
- The IAMs are limited in their spatial resolution, with some elements represented at national/regional level and others at grid cell level. The former becomes an issue when considering, for example, non-$CO_2$ GHG (including particulates) emissions and certain impacts. Some of this might be ameliorated with good downscaling of these elements

**Specific Emissions Pathways Recommended**
The specific REPs proposed are reasonable, but I do have a couple of concerns.

- In describing the pathways, e.g., in Table 1, the authors need to be clear in the names and the key characteristics that they are meant to be illustrative of ways that the emission pathways might come about. Specifically, the current names and descriptions of the key characteristics include explicit assumptions about socio-economic, technological, and policy elements. This is at odds with their desire that "the REPs should remain separated from the underlying socio-economic scenarios."
- None of the pathways deal with the possibility of the use of SRM, which would present some of the same, but at the same time quite different, challenges for ESMs and (AO)GCMs as pathways with net negative emissions at some point in the future. It has been argued elsewhere that SRM is not currently part of the climate change policy debate, but this is increasingly inaccurate and likely to be even more so over the time frame over which these REPs are meant to be applied. Not including some consideration of SRM would set up the community to be in a position of doing a lot of catching up in the future, much as has been argued about the need to do a better job of dealing with CDR at this point.

**A Final Comment**

I enjoyed reading this paper and feel that it is an important contribution to the further development of scenarios for CMIP7 and AR7. I look forward to the final version.

**References**

Masson-Delmotte, V., Pörtner, H.-O., Roberts, D. C., Shukla, P. R., Skea, J., Zhai, P., Cheung, W., Fuglestvedt, J., Garg, A., O'Neill, B., Pereira, J., Portugal Pereira, J., Riahi, K., Sörensson, A., Tebaldi, C., Totin, E., van Vuuren, D., Zommers, Z., Al Khourdajie, A., Connors, S. L., Fradera, R., Ludden, C., McCollum, D., Mintenbeck, K., Pathak, M., Pirani, A., Poloczanska, E. S., Some, S., and Tignor, M.: Workshop Report of the Intergovernmental Panel on Climate Change Workshop on the Use of Scenarios in the Sixth Assessment Report and Subsequent Assessments, Working Group III Technical Support Unit, Imperial College London, United Kingdom, 67, 2023.

van Vuuren, D., Tebaldi, C., O'Neill, B. C., SSC, S., and participants, w.: Pathways to next generation scenarios for CMIP7: ScenarioMIP workshop report. 2023.

---

## Author Response (AR1)

**6 March 2024**

**Table of contents**

**Author replies to RC1**

We thank the reviewer Andy Reisinger for his in-depth review. Please find our replies inline below in blue.

GENERAL COMMENTS

RC1.1: The perspective by Meinshausen et al proposes a set of representative emission pathways (REPs) to be used to drive the next set of Earth System Model experiments, with a view to provide a core set of climate framing pathways for the IPCC 7th Assessment Cycle. The perspective is well written and makes a cogent argument supported by a set of policy- and science-related criteria, having drawn on a wide and diverse range of authors.

REPLY1.1: Thank you.

RC1.2: I have only one high-level concern, which is whether the manuscript sufficiently differentiates the broader narrative value of REPs from the narrower question of what REPs should be used to run ESMs. I.e. it is one question what REPs would best serve as climate framing pathways to support discussions under the UNFCCC and Paris Agreement/GST2, to provide a range of climate futures for the next IPCC assessment cycle, and to guide research across a broad range of domains; it is another question whether all those REPs necessarily need to be run specifically through ESMs (as a resource-intensive community effort) to produce the scientific climate knowledge that those broader processes need. As noted in a community comment by Robert Kopp, some of the differences in impacts and risks in the half-degree space between 1.5 and 2 degrees (or between 'somewhere below' and 'slightly above' 2 degrees under different confidence levels in the achievement of current long-term targets) are not

necessarily best resolved by ESMs but rather by more specialised models that either probe a specific biogeophysical domain (such as ice sheet responses to warming) or the interaction between human, ecological and biogeophysical drivers of impacts and risks.

REPLY1.2: Thank you. Indeed that is a useful consideration, which we pondered about. And in addition to the UNFCCC, Paris Agreement, the IPCC cycle, as well as the 'pure' scientific inquiry, a broad spectrum of user communities depends on the outputs (or chain of analysis) that themselves depend on these foundational ESM runs, such as the user communities related to physical risk assessments etc.. Hence, it is indeed important to strike a good balance.

The same discussion on the right balance has in some form or another preceded the selection of each of the previous SRES, RCP, and SSP design cycles, with the ex-post decision in IPCC assessments on which scenarios to focus on. History has shown that (for legitimate reasons), the scenarios that are not run by ESMs do not have the capability to serve as cross-community scenarios to investigate e.g. the impact of higher or lower overshoot within the Paris Agreement target range. As mentioned in our earlier reply, the lack of sufficient granularity in ESMs runs leaves WGII no choice but to resort to rather general statements about "overshoot" without being able to investigate the differences (or lack thereof) that a 1.7 or 1.8 or 1.9°C overshoot of the 1.5°C global warming level makes. In our mind, the time has come to focus some of the resources on providing the foundation for the long chain of science and analysis that is needed to investigate this question of utmost policy relevance as it makes a trillion dollar difference in terms of the implied speed of the energy transition (e.g., in terms of stranded assets). Previous attempts in AR4 to cover the lower scenario range with a hypothetical 'concentration stabilisation' scenario, in AR5 to extrapolate from RCP2.6 towards 1.5°C compatible scenarios or AR6 to investigate overshoot with a more pronounced overshoot scenarios (like SSP5-3.4-OS) did demonstrably not result in the chain of analysis that is required to adequately inform decision makers on the differences (or lack thereof) of emission scenarios that imply 5-to 10 years earlier or later emission phase-outs, even though that is the granularity of decisions that are being discussed today on the mitigation side. While pattern-scaling, stitching and statistical emulation techniques have advanced considerably, the universal applicability, robustness and scientific standing are far inferior to original ESM runs that could precipitate the analysis chain of subsequent impact, adaptation and vulnerability studies. Thus, while our proposal keeps the overall number of proposed scenarios limited to the same as or less than previous rounds, we purposefully put some higher emphasis on the future emission spectrum that can serve a range of user communities, including mitigation and adaptation decision makers.

One more note on the computation burden for modeling centers to provide some perspective on the argument that similar scenarios (such as e.g. a 1.5°C low and a higher overshoot scenario)

cannot be afforded. We here reproduce the Table from page 9 from the CMIP7 design document[1].

| Activity | Coupled | Atmosphere only | Land only | Activity total |
|---|---|---|---|---|
| DECK[1] | 1475 | 136 | | 1611 |
| ScenarioMIP | 695 | | | 695 |
| AerChemMIP | 1400[2] | 495 | | 1895 |
| C4MIP | 900 | | | 900 |
| CFMIP | 300 | 201 | | 501 |
| DAMIP | 1620[3] | | | 1620 |
| GeoMIP | 50 | | | 50 |
| LMIP | | | 175 | 175 |
| PMIP | 100 | | | 100 |
| RFMIP | | 456 | | 456 |
| **Grand Total** | **6540** | **1288** | **175** | **8003** |

The table shows that in terms of overall model years, the future scenarios are not insignificant but by far not the highest computational burden on modeling centers participating in CMIP. A total of 1475 model years in the coupled model versions have to be run before any other experiments can be performed. With a total of an estimated 6540 model years (assuming single ensembles), future scenarios will only consume 10% of the computational burden. Thus, one more or less scenario would consume roughly 2% for modeling centers that intend to participate in all of the key CMIP activities (again, assuming single ensembles).

RC1.3: This does raise the question whether the resources needed to run all REPs through ESMs as a community effort are indeed spent well enough, and to what extent a (slightly) smaller or different set of REPs might be equally justifiable as far as ESM runs are concerned, while treating it as a separate, related but distinct, question what REPs should be used to frame alternative climate futures more generally to support climate policy processes in the UNFCCC and IPCC and to drive broader research efforts across multiple domains.

REPLY1.3: Thank you. For the purposes of this paper, we discuss the REPs as the REPs that are intended as pathways for ESM experiments. We attempted to provide an indication of potential 'priority' runs, that would allow ESM groups with limited resources to prioritize certain runs. They are indicated in Table 1, with higher priority scenarios being four scenarios.

- 'NFA' (No further action) reference scenario,
* * *
[1] CMIP (2024) "Community consultation: CMIP AR7 Fast Track v2", available at: https://wcrp-cmip.org/wp-content/uploads/2024/02/v2_supporting-information_Open-community_FINAL.pdf.

- 'DASMT' (delayed action and stabilization, but missing target),
- 'DAPD' (delayed action and peak and decline)
- 'IAPD' (Immediate action and peak and decline)

For the CMIP6 scenarios and IPCC AR6, there have been effectively five 'Tier 1' high priority scenarios, i.e. the four identified by ScenarioMIP (SSP1-2.6, SSP2-4.5, SSP4-6.0 and SSP5-8.5), in addition to the 1.5°C compatible scenario SSP1-1.9 (that has not been selected as 'high priority' by ScenarioMIP), but run by a high number of ESMs and used as high priority scenario in IPCC AR6 for obvious reasons. Thus, our proposal effectively reduces the number of 'high priority' runs required by ESM from five to four to address the concern about resource requirements by these ESM runs.

RC1.4: However, I accept that this cannot be answered based solely on the unique scientific knowledge that the ESM runs would provide. ESM runs have signalling and narrative power. E.g. even within the IPCC, ESM-run scenarios will very likely pre-structure the range of alternate climate futures within which the assessment across WGII and WGIII (or within any Special Report) occurs. Any REP that is not covered by ESMs risks playing a lesser role not only in the WGI domain but also in the assessment of differences in risks, and of adaptation and mitigation responses considered by WGII and WGIII in the next IPCC assessment – or at least any scenario not run through ESMs would have to work much harder for its justification in the overall narrative, including in the IPCC approval process. This means a strong, but separate(!), argument can be made to run an emission pathway through an ESM even if it was not strictly necessary from a scientific perspective to have that scenario modelled through an ESM. The authors set up criteria from both policy and science domains to argue for their selection of REPs, but there is no discussion of whether the policy and science criteria may in some cases lead to contradictory conclusions about a preferred set of REPs; and if so, based on what considerations the authors reach their final recommendation.

REPLY1.4: Thank you for this most pertinent point and perspective which we wholeheartedly agree with (i.e. the strong signaling and narrative power of ESM scenarios). We would not characterize it as much as "policy and science criteria may in some cases lead to contradictory conclusions" as rather "policy and science criteria imply a different emphasis". The most obvious area of relevance is the spacing between the stronger mitigation scenarios (see discussions above). Given the strong policy interest and science's inability so far to provide detailed answers (as exemplified by the under-representation of specific information in WGII on relevant 1.5°C or 2°C overshoot scenarios, simply because the relevant ESM scenarios are not available to drive the impact studies), we would argue that a strong policy interest is also triggering policy-relevant science questions. Scientifically, the question about detection and attribution with less pronounced signals can pose quite a challenge, but makes for intriguing research questions. For example, detection of climate change in a single day of weather around the globe would have been perceived to be swamped by noise, yet new techniques are able to delineate between the

climate change signal and day-to-day weather variability (Sippel et al., 2020, https://www.nature.com/articles/s41558-019-0666-7).

The ESM runs have scientific robustness that cannot be replaced with currently available emulator techniques. As a result, we feel that it is worth investing extra resources into these runs because of the critical need for such information as the world tries to grapple with the differences between reaching net zero in 2040 vs. 2050 vs. 2060. We think that these extra resources are worth spending because they have a comparatively relatively small marginal cost (see discussion above under RC1.2) and even if there is a risk that the difference between the runs is swamped by natural variability. We expect that significant resources will continue to be invested into other ESM-based analysis in other MIPs, which generally have a wider scientific focus than the specific case of future scenarios. As we said, we feel that the ESMs have a unique role to play in the scenario space and this cannot be replaced by any other currently available approach.

In summary, we argue that interesting science questions are to be found across the whole spectrum of scenarios, different gas contributions, different regional emissions, different timing of emissions, and we have outlined several of them. Policy-relevant questions are however more confined. Thus, rather than portraying science and policy fields of interest as 'contradictory', we would like to argue that the advancement of scientific inquiry will need additional experiments beyond policy-relevant scenarios that are covered in the specialized MIPs, investigating hypothetical scenarios, abrupt transitions, long integrations, individual forcer variations, etc. In this context, we suggest to add a clarifying paragraph at the end of section 3:

> *"The impression could arise that the policy and scientific objectives relevant to the framing pathways are in conflict. While a policy-relevant question almost always entails a scientific question of interest, the scientific realm of questions is broader. For example, the policy interest in the differences, in terms of impacts, between pathways with low and medium overshoot of 1.5°C scenarios also includes interesting scientific challenges. For example, how to quantify, in more detail, the IPCC finding that 'every bit of warming matters', using for example new statistical techniques to detect climate change signals (e.g. Sippel et al., 2020). Staying with this example, previous designs of the framing pathways did not provide the opportunity to investigate the extent to which we can detect signals that might at first sight be considered too small given the size of natural variability."*

Also, we suggest to insert one sentence in the beginning of section 3, highlighting the specialized MIPs that are the main driver for scientific advances:

> *"Thus, many of the scientific advances can be expected to emanate from the specialised MIPs (CFMIP, HighResMIP, AerChemMIP, C4MIP, RFMIP, CDRMIP, GeoMIP, LUMIP, ISMIP,*

*OMIP, VolMIP, DAMIP etc - see https://wcrp-cmip.org/model-intercomparison-projects-mips/)*
*that are conducted in parallel to running ESMs with the multi-gas scenarios. "*

RC1.5: So my overall sense is that the manuscript could work harder to disentangle and disclose the different motivations and arguments relating to those different objectives, and the degree to which different REP choices and different modelling tools using those REPs might best respond to the authors' set of criteria, and to reflect more critically and transparently on whether there might be a difference between the proposed set of REPs to serve as broad climate framing pathways in general, and REPs that are run specifically through an ESM community effort. Many of my specific comments point to areas where I think this clarity could be increased. But in the end, I fully accept that the balancing of those considerations does rely on judgment, meaning that reasonable people can look at the same set of facts and nonetheless come to different conclusions.

REPLY1.5: Thank you for these very helpful and specific suggestions, which we address in the comments and our replies below.

RC1.6: Despite my multiple (exhaustive but hopefully not exhausting) specific comments below, I'm happy to regard this as a minor revision, since I consider the manuscript to be already of high value - but there's an opportunity for a clearer discussion of those differences to lift its value and utility further.

REPLY1.6: Thank you. Much appreciated.

SPECIFIC COMMENTS

RC1.7: L77-78: this sentence condenses a broader use (use of climate framing pathways in the IPCC AR7) and the specific evaluation of those pathways through CMIP/ESMs; consistent with my general comment, I wonder if authors could try to recognise more that there may be a distinction between those two uses, and the implications of that.

REPLY1.7: We note that. Without going into great detail we adapt the specific wording that currently reads "*for the next generation of framing pathways that is being advanced under the CMIP umbrella for use in the IPCC AR7*". That wording can indeed be read as CMIP and IPCC AR7 purposes are one and the same. Thus, the adapted wording now reads:
*"for the next generation of framing pathways that is being advanced under the CMIP umbrella which will influence or even predicate the IPCC AR7 consideration of scenarios*"

RC1.8: L94: "well before 2028" in my view underplays the urgency, set out in lines 746ff. Suggest consider slightly rephrasing this.

REPLY1.8: Understood. It now reads "*by 2026 or well before 2028*".

RC1.9: L101-102: framing pathways can shape how the next IPCC assessment approaches alternative climate futures in general; or they can simply mean "how ESMs can be run with a consistent set of drivers". I suggest authors disentangle these very different meanings of 'framing' in different places of the manuscript. L113-114 acknowledges this point, but I don't see this distinction elaborated in the manuscript and how different objectives (the broad and the narrow use of 'framing') might lead to different conclusions. Sections 5.2 and 5.6 to some extent speak to this issue, but these sections come across as defending the authors' REP choice rather than working through the different implications and disclosing judgements that lead authors to reach their conclusions.

REPLY1.9: Thanks for that comment, although there might be a slight misunderstanding in our wording "how ESMs can be run with a consistent set of drivers", as our sentence continues after the brackets to read "to build a range of climate futures which in turn provide a common framing input to impact and vulnerability studies can be conducted ". Thus, we do not consider any pathways with a "consistent set of drivers" as being a "framing pathway". Framing pathways are those that cross Chapter and Working Groups (to stay within the IPCC context for a moment) and hence provide one (but not the only) "dimension of integration". We also hoped that the following sentence " The framing pathways thereby can provide a backbone of integration across the IPCC physical science (Working Group I) and impact (WG II) communities and also to link with socio-economic and mitigation information (WG III) " makes clear that the framing pathways can be strongly influencing the IPCC framing, although those are not the exclusive approach to achieve a tie across chapter and working groups.

Anyway, on the point of distinguishing between two types of 'framing pathways', we feel that this might have arisen from a not ideally worded section. We hence suggest to modify the sentence in question to change from:

> "Such are hereafter referred to as framing pathways since they frame how ESMs can be run with a consistent set of drivers (emissions, concentrations, land surface states, solar activity etc.) to build a range of climate futures which in turn provide a common framing input to impact and vulnerability studies can be conducted (Frieler et al., 2023; Warszawski et al., 2014)"

To now read:

*"Such  (Frieler et al., 2023; Warszawski et al., 2014)"*

RC1.9: L131-132: this might be worth elaborating a bit: what is meant by the IPCC mandate here? What is the role of ESM runs in setting up an envelope of climate futures more broadly (beyond the immediate value-add from the specific knowledge gained)? Do ESM runs hold a special role in defining this envelope not (just) because of the knowledge they provide but perhaps also because they will be seen as unassailable cornerstone in the intersection between science and policy?

REPLY1.9: Thank you. We now clarified the 'IPCC mandate' as being specifically (the mantra of) 'policy-relevance but not policy prescriptiveness' and added another sentence that refers to the implicit 'power' of scenarios of framing and determining future knowledge generation. The adapted section now reads:

*"Such use cases need to be considered when designing a new framing scenario set. The assessment of scenario-based information is central to the IPCC in particular to provide climate information that is societally and policy relevant, but not policy prescriptive. To support the IPCC in fulfilling this mandate, we argue that it is important that the scenarios run by ESMs cover a wide range of policy or physically relevant futures. That is mainly due to the unique position that ESMs play in IPCC assessment reports. They determine the boundaries of scenario exploration across various research communities hence the choice of pathways to run with ESMs is particularly crucial (because, by implication, any scenarios not covered by an ESM simulation will receive little, if any, attention). "*

RC1.10: L136ff: I find the distinction between pathways and scenarios (especially as stated in lines 141-145) contradictory and confusing; line 143 says that pathways do not provide any explicit assumptions about socio-economics or policy, but then lines 143/144 say that pathways may describe quantified socio-economic futures. My interpretation is that pathways tend to be more uni-dimensional, whereas scenarios describe a complex set of interconnected and internally consistent drivers, assumptions and outputs. So pathways can cover any type of variable (input or output) that can be extracted from scenarios – emission pathways, temperature pathways, perhaps aggregated into climate pathways, but also energy-specific pathways, or socio-economic pathways. So it's not whether socio-economics is in or out, it's the uni- (or simply narrow) dimensionality of pathways compared to the multi-dimensional and internally consistent set of assumptions, drivers and outputs that characterise scenarios. I'm not

saying the authors have to adopt this way of distinguishing pathways and scenarios, it's just a suggestion that may be consistent with what the authors had in mind but the current text does not get this across. Box 1 is slightly less confusing, but it could also be clearer about the distinction between pathway vs scenario.

REPLY1.10: We very much appreciate this call for clarity. And yes, the reviewer is very much correct that we propose (in line with earlier literature) to use the 'pathways' as the uni-dimensional or more narrowly scoped depiction of a future trajectory, whereas, as the reviewer says, scenarios describe the 'complex set of interconnected and internally consistent drivers, assumptions and outputs'. We hence modified line 143 to read instead of:

" *Building on the definitional distinction in van Vuuren et al. (2014), we focus here on 'pathways' that describe a climate-related transient evolution of the future (emissions, concentrations, geophysical climate), without any explicit assumptions about socio-economics or policy.* "

Now as:
" *Building on the definitional distinction in van Vuuren et al. (2014), we focus here on 'pathways' that* **tend to be more uni-dimensional or narrow descriptions of a potential future, for example,** *climate-related transient evolution of the future (emissions, concentrations, geophysical climate), without any explicit assumptions about socio-economics or policy.* " *- the text then continues as before to contrast pathways with scenarios.*

RC1.11: L158ff: this section could do more to acknowledge the overlap between climate and ecosystem/land-use models, which can fall into the gap between 'climate' and 'socio-economic' domain. These are arguably becoming more important, especially in the context of overshoot and land demand for CDR. Also somewhere in this section might be the place to more explicitly and in more detail acknowledge and map out the space that more detailed and specialised biogeophysical models can and must play to complement ESM runs to understand the difference that half a degree could make for some key dimensions of climate-related risks – and the degree to which such models must have ESM runs as their input, or whether they could also do their job using other inputs that are less resource-intensive to produce. This is critical to the question whether the set of REPs proposed as 'framing' pathways in a broader sense necessarily and in its entirety must be run through ESMs.

REPLY1.12: This is indeed a key point. Firstly, we strongly agree with the emergent importance of biogeophysical impact, ecosystem and land-use models. In our assessment, there are two

factors influencing whether those models are run with scenarios other than those run by the ESMs.

Firstly, whether an alternative approach can provide the same data and its internal integrity. That means that just having disjunct surface temperature and precipitation projections, even if the statistical properties of both timeseries are perfectly replicating those of ESMs are only of use, if also the joint probabilities are appropriately reflected. Some approaches also might either ignore larger spatial correlations (which would then for example prohibit a study on concurrent droughts in the world's bread baskets, for example) and some studies are not suited for any impacts that depend on integrated variables (e.g. for glacier runoff, some hydrological studies, and sea level). The importance of the internal data integrity of course applies to the full range of variables that the impact/ecosystem/land-use models use from the ESMs, such as cloud fractions, surface shortwave downward fluxes, wind speeds etc.. To our knowledge, there are multiple partial solutions to providing ESM emulators (e.g. MESMER, STITCHES, etc.) but there are to date no emulators available that can preserve statistical properties across the multi-dimensional variable space and full spatial domain.

In addition, there is no guarantee that these emulators can adequately distinguish climate state for a given warming level before and after a temperature overshoot because most of the underlying hypothesis are based on pattern/temperature scaling approach without accounting for the legacy of the warming seen before.

Also, it is important to recall that emulators are only "as good" as the data they're trained on. And in the absence of sufficient training data on overshoots, they are actually not suited to explore research questions in that space. Most notably, prominent examples such as MESMER and STITCHES rely essentially on linear pattern scaling with GMT to represent the forced signal. This is well established for scenarios of continuous warming, but much less so beyond the peak - neither long-term changes under stabilisation (i.e. King et al. 2020) or after overshoot (i.e. Pfleiderer et al. 2024) can be assessed using such tools. So they'd be arguably better suited to explore the difference between continuous warming scenarios between 2 and 4-6°C than to represent the differences between high and low overshoot pathways.

The second question is whether any of the approaches that solve those issues and that might emerge in the future would have the scientific standing that would then serve as an equivalent approach to justify its use in the a) the biogeophysical impact / ecosystem and impact models and then also be equivalently represented in the follow-up assessments (such as IPCC AR7). From historical experience, the empirical evidence in IPCC report writing processes seems to be that non-tier 1 ESM scenarios have a strong hurdle to surpass to be included as high priority scenarios - with the exception of a 1.5°C degree compatible scenario that was absent from the last ScenarioMIP Tier 1 recommendation set for example, given the strong dissonance with the political discussion that focussed on 1.5°C degree. However, a scenario that would not even have been run by ESM scenarios seems rather unlikely to have the required standing in the

community to be considered for high-level IPCC assessment of the literature, partly because the evidence base will be comparatively thin.

Anyway, we fully agree with the reviewer on the emergent importance of these biogeophysical impact, ecosystem and land-use models. However, we consider this section to be less ideal to draw attention to this emergent importance as it mainly deals with the history of the matrix approach. We think that highlighting ecosystem and land-use models explicitly in our Figure 1 and adapted section 5.2 accordingly to read now (bold is added material):

> *"Similarly, we argue that it is time to enable WG II to provide the corresponding impact assessment via adequate ESM simulations that provide the needed geophysical input* **for ecosystem, land-use and biogeophysical impact models and**  *associated impact studies.* **Those types of models that directly feed off ESM output have an emergent importance in the quest to arrive at a finer-grained picture of future impacts and their difference between different pathways.** *"*

RC1.13: Figure 1: As noted in a community comment by Alex Magnan, there is no linear path from REPs to RIPs and RAPs, or between RTPs, RSPs and RAPs. It may not be worth spending too much time re-doing the figure, but if the authors can think of a way to represent the cluster of pathways on the left hand side as a dynamically interconnected set of issues rather than a linear progression from one to the other, that might be very useful (since the figure may well be used in lots of presentations to show the overall REP and scenario logic). It may also provide an opportunity to highlight the much smaller geographical scale, and dependence on nuanced socio-economic assumptions (including e.g. the potential for different regional-/national-level SPAs nested within a single global-scale SSP that will then drive diverse regional-/national-scale risks), that will be relevant for RIRTs, RAPs, and RTPs.

REPLY1.13: A very valuable suggestion. We hesitate to make Figure 1 more complex, but considered a graphical language that can make it more intuitively clear that there is a separate multi-dimensional space for each of the policy, socio-economic and adaptation domains that is somewhat orthogonal to each other - unlike the emission, concentration and warming pathways on the right side. Thus, the right side is visually not connected by these fanning areas of uncertainty, whereas the left side consists more of the multiple 'popcorn' heaps. We hope that this visual adjustments implement that very good point that both reviewers made.

BTW: We left the fanning uncertainty connection from RSPs and RTPs towards REPs, as there is a somewhat deterministic connection between the three. Anyway, we hope this below version is considered to be a useful improvement.

[Figure]

RC1.14: L298ff: I don't disagree with the points made in this section, but it's not clear whether all those issues rely on a community ESM modelling effort. The knowledge needs related to many of those questions rely on modelling approaches very different to ESMs (and, I would argue, in some cases not at all dependent on ESMs). On substance, I would add a criterion relating to the increasing need to consider overshoot pathways from a policy perspective (i.e. how to achieve a

decline in temperature after a peak has been reached, and the extent to which this ought to influence near-term policies when temperatures are still on an upward trend).

REPLY1.14: We agree with the reviewer that in addition to ESM runs, other modelling approaches would need to complement ESM outputs to address those questions. However, often even the alternative approaches (such as calibrated reduced-complexity models) will require an enhanced confidence that those techniques adequately represent system dynamics for policy-relevant scenarios (such as appropriately distinguishing between low and high overshoot scenarios).

On the substantive point of overshoot, we amended the existing overshoot point (point h) accordingly, so that two new sentences at the end are now:

> *"This would also enable policy-relevant information in what way envisaging declining future temperatures might influence policy choices today, while temperatures are still rising. Any co-benefits and more ancillary impacts of future negative emissions options (beyond land based carbon dioxide removal) require an enhanced focus on multiple options to reach 1.5°C given that current temperatures are already approaching this level. "*

RC1.15: L325-333: A couple of questions: it's not clear why high-end global warming outcomes are of particular importance for a loss and damage conversation, given that loss and damage (as far as I understand) relates to actual losses and damages, not projected risks? Secondly, precisely because the range of near-term warming is dominated by climate uncertainties rather than differences between emission scenarios, this seems to argue against placing too much emphasis on a wide range of REPs if the goal is to inform loss and damage conversations and near-term adaptation needs? I don't disagree with the overall thrust of the discussion in this section overall, but the arguments in those lines don't seem to stack up well.

REPLY1.15: Thank you for this comment. We realize that the early mention of 'loss and damage' might be confusing as it indeed seems to suggest that loss and damage is intrinsically linked to the high-warming scenarios. The middle of the paragraph actually clarifies our argument of why the full range of pathways will be important for the loss and damage discussions. From below (which I think the reviewer fully agrees with): to determine the unavoidable amount of loss and damage. From above: as loss and damage often relates to the extreme and tail ends of the distribution, we argue that a lack of large ensemble sizes of some ESM (in order to explore the tail ends of the climate extremes) can be partially addressed by following a global warming level approach, in which higher-scenario and later-in-the-century segments might be transferred to indicate end-of-high-tail warming outcomes in the near term. In that way, the higher scenarios could indeed then be of importance for projecting potential high-end-tail loss and damage

potentials in the near-term. Anyway, we hope that deleting 'loss and damage' from early on in the discussion does not mislead the reader and offers our line of argumentation first to avoid misunderstandings.

RC1.16: L333-339: these points are useful but are left hanging. Given that the authors ultimately recommend including something like SSP3-7.0 in the REPs, I'm missing a conclusion that says something like "on balance, we feel that because of small ensemble sizes, we will learn more if we use the median results from SSP3-7.0 and treat those as proxy (potentially with scaling) for high-end climate outcomes under current policies". Without clarity about why SSP3-7.0 should be included, SSP3-7.0 might just become the new RCP8.5 for future arguments about the validity and utility of high-end emission scenarios. Of course there is also the separate rationale for SSP3-7.0 being the "world avoided", but is that the main reason to include it, or is it (also) because it can and should be used as a proxy for high-end warming under current policies? Or is it also because we will learn something about earth system responses to climate forcing that is important to learn, regardless of its relevance from a narrative space?

REPLY1.16: Thank you. The reviewer nicely summarizes the three main reasons for including the 'world-avoided' scenario SSP3-7.0.

a ) providing the backdrop of 'the world avoided' as an approximate illustration of how changed economics in the mitigation space changed the needle from SSP3-7.0 or SSP5-8.5 (arguably, the needles have never been up there) towards a 'current policy' scenario.

b) populating the high-warming space (that can be useful to examine high-end-tail distributions of 'current policy scenarios) more efficiently, as otherwise very high emission versions of 'current policy scenarios would be needed.

c) learning about geophysical climate system responses more theoretically, if one 'pushes the system hard' (although this is arguably better examined in dedicated MIPs/experiments, e.g. abrupt-4xCO2, rather than trying to have it as a side-effect of a scenario).

We attempt to avoid too many redundancies in the manuscript, which is why we hope that the tabular overview in Table 1 provides a succinct summary of these three rationales (in addition to a fourth one). In that column 'advantages', the reason for the 'the world avoided' scenario is summarized as:

- *Allows depiction of the world that could have unfolded without climate policies.*
- *Provides insights into high tail warming possibilities of lower emissions scenarios.*
- *Allows direct comparison of new generation of ESMs with older ESMs, if a CMIP6 high end pathway is repeated (SSP5-8.5 or SSP3-7.0).*
- *High signal-to-noise for projected changes in climate to learn about climate system properties, if the system is 'pushed hard'.*

RC1.17: L346ff: these policy criteria are all useful, but it is not clear to what extent they rely on ESMs to provide the answers that policymakers need. A critical reflection on differences (where they exist) between policy/narrative value, and scientific value, of different REPs would in my view significantly lift the value of this manuscript for the scientific community processes that ultimately have to reach a decision on REPs for ESMs.

REPLY1.17: Thank you. As we mentioned above in our reply RC1.4, we view these potential differences as nearly one-sided, in the sense that every policy-relevant question (if not already answered by science) almost always contains scientifically interesting questions, whereas not all scientifically interesting questions entail direct policy-relevance. Many of the scientific advances are expected to be achieved through the specialized MIPs, i.e. LUMIP, C4MIP, RFMIP, PMIP etc.. Hence, as noted above, we added the following text in section 3:

> *"Thus, many of the scientific advances can be expected to emanate from the specialised MIPs (CFMIP, HighResMIP, AerChemMIP, C4MIP, RFMIP, CDRMIP, GeoMIP, LUMIP, ISMIP, OMIP, VolMIP, DAMIP etc - see https://wcrp-cmip.org/model-intercomparison-projects-mips/) that are conducted in parallel to running ESMs with the multi-gas scenarios. "*

> *We argue that striking a balance between characteristics that are of relevance to the science (section 3) and to policy (section 2) is less of an intricate problem than it first appears. For one, in most policy-relevant questions, there is a science question behind it (unless it has already been answered). For example, there is high policy relevance in having a high granularity of pathways that have - generally speaking - 'close proximity', but which explore the policy-relevant target realm between 1.5°C and well below 2°C. From a scientific perspective, the delineation between two 'close' pathways, detecting and attributing the 'differential' biogeophysical and socio-economic impacts is a challenging, but interesting scientific question that has received only limited attention to date. The second reason, why the choice of scenarios is less in need of getting 'the balance' right between scientific and policy questions (however stakeholders might observe them) is that most of the scientific advances are for example gained from hypothetical, abrupt, or single-forcing experiments, which are covered in the specialised MIPs.*

RC1.18: L346ff: From a policy perspective, one point missing is the question of the extent to which there is flexibility in treating non-CO2 mitigation differently to CO2 mitigation, within a given climate goal and over different time scales. Existing emission scenarios in the "below 2°C" space tend to assume a similar stringency for non-CO2 mitigation as for CO2, and hence both SSP1-1.9 and SSP1-2.6 have rapid and deep reductions of CH4 (and other air pollutants), well beyond simply co-mitigation from reduced fossil fuel use. But that's not what we're seeing in the policy world, where non-CO2 policies tend to be constructed very differently and with much less stringency. So from a policy perspective it would be useful to probe the feasibility and climate consequences of a pathway where governments do meet net-zero goals for CO2, but don't apply the same stringency to non-CO2 mitigation as is implied in current scenarios.

REPLY1.18: Thank you. We very much agree with this point. We feel though that the general point (f) on 'Non-CO2: What are the effects of non-CO2 mitigation?' is a good place to capture this aspect. While the 'feasibility' questions from a mitigation action point of view are less relevant per se for the ESM runs (but of course play a role in the scenario derivation), we added another sentence with regard to the climate effects in general:

> ""*Generally, examining the flexibility governments have to focus more or less on CO2 versus non-CO2 greenhouse gases and air pollutants in terms of climate outcome consequences is of interest - aiming for a more encompassing reflection of non-CO2 atmospheric interactions and uncertainties. This latter aim could be linked to the emerging capability of some ESMs which have greenhouse gases to be emissions-driven is emerging in some ESMs (Folberth et al., 2022).*"

RC1.19: L417:422: I would have thought that a more fundamental question is to what extent is climate change itself reversible? That's what ESMs answer. The reversibility of impacts and risks needs much more complex scenarios and models that inevitably depend on socio-economic assumptions (and/or ecosystem models that may not rely on ESMs as input to answer questions about reversibility). From a policy perspective, I would also add questions about whether an intention to achieve net negative emissions in future will/should affect the way that we approach mitigation in the near term?

REPLY1.19: Thank you. We now adapted the headline question for this point to read "To what extent is climate change and the impacts of climate change reversible and on what timescales?", i.e. adding climate change itself. We had this differentiation between climate change and climate change impacts in the second last sentence, but hopefully bringing this up is adding clarity. On the last question, and as we mentioned above, we now added a segment to this bullet, saying:

> "*This would also provide policy-relevant information on the question of the extent to which envisaged declines in future temperatures might influence policy choices today*

*(when temperatures are still rising). Any co-benefits and more ancillary impacts of future negative emissions options (beyond land based carbon dioxide removal) require an enhanced focus on multiple options to reach 1.5°C given that current temperatures are already approaching this level."*

RC1.20: L433: (and L45): given the general push of the authors to look beyond 2100, I found the distinction between "up to 2100" and "beyond 2100" a bit jarring here. "Beyond 2100" to me does not signal multi-century scales relating to sea level rise, which is a key (science and policy) question.

REPLY1.21: Thank you. From the point 'j' we deleted the timeframe as it is not central to the point "What could the worst-case outcome world look like?". Further below, in point 'm', i.e. what the long-term implications are beyond 2100, we just mention 2100 as the historical cut-off point for most studies. The multi-century scales are explicitly mentioned. We hope that this clarifies that we are not particularly looking for an analysis 'up to 2100' and 'beyond 2100'.

RC1.22: L470ff: One point missing from this list, in my view, is better representation and intercomparison of the role of non-CO2 emissions in driving climate and uncertainties (see also L544), especially for CH4. How well do we really understand the feedbacks that influence CH4 chemistry in a warming climate? One can of course work around this by driving ESMs with prescribed concentrations, just so that more models can run the full set of forcings, but this potentially hides a non-trivial element of uncertainty. More specific results might also help calibrate emulators better for climate responses to CH4 emissions. So progressing this space would strike me as a highly relevant science question.

REPLY1.22: We think that this point can be very well reflected in our long bullet f on non-CO2 forcers. The last two sentences that we added to emphasize this point are: *"Generally, examining the flexibility governments have to focus more or less on CO2 versus non-CO2 greenhouse gases and air pollutants in terms of climate outcome consequences is of interest - aiming for a more encompassing reflection of non-CO2 atmospheric interactions and uncertainties. This latter aim could be linked to the emerging capability of some ESMs which have increasingly enhanced capabilities to integrate a broader range of atmospheric variables and complexities. Furthermore, the potential for non-CO2 GHGs greenhouse gases to be emissions-driven is emerging in some ESMs (Folberth et al., 2022)."* (see above our reply to RC1.18.)

RC1.23: L471-478: I agree with the point being made here, but a single scenario will not help us clarify this. What we would need are two scenarios with similar levels of CO2 mitigation, and then different efforts to reduce non-CO2 emissions. This suggests that answering this sort of question requires a different emission pathway architecture, but might then also require more

limited ESM intercomparisons, or the use of other models. So it's not clear how this point relates to the goal of this manuscript, which is to propose a set of REPs to be used to drive ESMs as a community effort.

REPLY1.23: We fully agree with the reviewer, that this point alone would ideally follow a very specific scenario architecture that considers a base scenario with several marginal variations for individual non-CO2 forcers and different timings and potentially also different regional schedules. However, we do not intend to confine the list of policy and science questions to the ones that are answered by our proposed set. Rather, we attempt to start with a broad horizon scanning of the policy and science questions that would ideally be answered by these framework scenarios. The proposed set is then one possible one that combines as many relevant aspects as possible without the aspiration of being the ideal set for every single sub-question - given the need to balance the policy and science question with available resources. Anyway, for this specific point, we added a sentence to clarify the ideal of multiple sensitivity scenarios for individual forcers. We still maintain that the 'emergence of mitigation benefits' can well be illustrated in a general way with the proposed set of scenarios - towards the previous reference scenario, i.e. against the 'world that would have been' scenario and one can also compare against the 'current policy' scenario to quantify the emergence and timing of climate change differences due to mitigation action in the lower scenarios. Hence, we added to the point (a), the following:

> *"Ideally, many individual sensitivity scenarios for individual forcers would be undertaken to investigate the emergence of climate effects due to mitigation action on individual forcers, but the overall framework design can assist in quantifying an aggregate effect of multi-gas mitigation action."*

RC1.24: L488ff: I would have thought that better understanding of feedbacks is relevant not just under overshoot but also for current policy scenarios? This would also specifically include risks to carbon stocks in terrestrial ecosystems, especially for carbon stocks that may have been enhanced deliberately as mitigation measure and claimed as removal to counterbalance continued emissions.

REPLY1.24: Very valid points indeed. We now changed the heading to include 'also under overshoot scenarios' so that it is not understood as an exclusive consideration of 'overshoot' scenarios. Furthermore, we added the very valid point of the 'reversal risk' for enhanced carbon storage, by adding *" as well as any risks to carbon stocks that were enhanced under CDR actions.".* Note however that in the below point (e) we also reference this issue when saying *"Global warming and related impacts on terrestrial ecosystems and their uses, particularly on their disturbances such as fires, drought, and pests (Westerling et al., 2006; Liu et al., 2023; Canadell et al., 2021) can also influence the durability of these CDR interventions in ways that are poorly represented in IAMs today."*

RC1.25: L498ff: I don't disagree with the question, but it's not clear that ESMs are best placed to resolve those differences in the half-degree space?

REPLY1.25: Given the high political relevance but also the lack of in-depth analysis with multiple ESMs, the ESMs are not the only tool to explore this question but ought to set up to enable the quantification of climate differences (and large ensemble sizes by some models, as have been submitted under CMIP6 will certainly help in getting a more fine-grained understanding of marginal differences) - as well as enable the considerations of differences by subsequent impact models. Even in the case that ESMs in any particular year would not show differences that stand out against natural variability, any long-term impacts that are dependent on the integral of climate changes would not be enabled, if ESMs once more do not provide the basis to investigate the half-degree or lesser differences. Thus, we argue that this is an important scientific gap with tremendous policy relevance (see also our other replies above on this point).

RC1.26: L506ff: I don't disagree with the question, but it's not clear that ESMs are best placed to answer those questions, given the crucial role of socio-economic conditions and policy design to understand the consequences of land-based CDR?

REPLY1.26: We fully agree that the socio-economic conditions are crucial and not part of the ESM remit. This paragraph only argues that ESMs are a necessary part of the mix of tools to fully investigate land-based CDR consequences.

RC1.27: L545ff: There is no linear logical way to get from criteria to specific scenarios, and I'm very happy to accept that the authors made (and had to make) some judgements to get from their criteria to this list. So I'm offering just some thoughts on other scenarios that could (in my view) equally have been in this list. I don't expect the authors to rebut those ideas one by one, it's more an invitation that if they have a clear reason why they did not choose one of those other scenarios, it might be worth including that rationale in the manuscript so that others benefit from their thinking.

In terms of comparability with CMIP6, I would see some value in having two backward compatible REPs. The obvious choice for this would be TEWA and IA2015 (SSP3-7.0 and SSP1-1.9), since both those scenarios could justifiably run with (by now) counterfactual emissions prior to 2023. This would provide a robust way to separate changes in model behaviour in the next generation of ESMs from changes in assumed future and past actual emissions.

> → REPLY 1.27a: Indeed a consideration worth exploring. In the end, it is a question of balance. The IA2015 scenario proposed here would have three characteristics that the SSP119 does not have: a) The IA2015 scenario would clearly follow the counter-factual of immediate emission reductions after the Paris Agreement 2015 reduction. With SSP119 being one of the higher scenarios until 2020, the difference would not be massive, but nevertheless not 'clean'. b) The IA2015 scenario would share the exact same set of historical harmonised emission and concentration histories

and hence serves as a clear reference scenario. Comparing a low CMIP7 mitigation scenario compared with SSP119 would still be hampered by (slightly) different historical emissions. c) The IA2015 scenario would share similarities with the IAPD scenario, in terms of gas-to-gas characteristics but also future negative CO2 deployment. Thus, the clean comparison of a delay in the onset of emission reductions is enabled, which is not a given with SSP119.

If one wants a clean comparison, CMIP could recommend to also re-run SSP1-19 with the new generation of models. The marginal cost is relatively low for a single ensemble and this avoids the headaches of close but not quite clean comparisons. A compromise solution doesn't seem the right choice to use, as it might just leave both questions (the hypothetical ambitious 1.5C scenario without overshoot starting directly after Paris with strong mitigation actions vs. a clean 'CMIP5 to CMIP6' model response comparison) not answered properly.

There is no emission pathway in this list that corresponds to a C3 category in IPCC WGIII. While I can see a narrative value of focusing on pathways that keep 1.5°C alive at least by 2100, it seems oddly dichotomous to characterise the lower end of the scenario space only by pathways that either miss "well below 2 degrees" entirely (because they end up at or above 2 degrees) or that do succeed in limiting warming to below 1.5°C in 2100. So I would regard a C3-type pathway as a highly policy relevant pathway. Not arguing that authors should add yet another 'close-by' REP to the list, but it might help the manuscript to recognise its absence and perhaps explain why the judgement was made not to have it.

→ REPLY 1.27b: Again, a very good point. In light of this, we added the following paragraph to the discussion of 5.2:

> *"An additional pathway that could be similar to the delayed action DAPD one could be one that limits peak warming at below 2°C but then avoids the strongly negative emissions. It would miss the 1.5°C warming level by 2100, and arguably would not be in line with Paris Agreement's 'pursuing efforts for 1.5°C' element, but could present a third plausible pathway within the 1.5°C to below 2°C range (similar to 'C3' category pathways investigated in IPCC AR6 WGIII). Given overall resource constraints, we consider the pair of IAPD and DAPD pathways to provide more relevant scientific and policy-relevant insights compared to either a IAPD - C3 or DAPD - C3 combination, yet acknowledge the advantage that a triple IAPD, DAPD and C3 investigation might bring. The reason we prioritise DAPD and IAPD is that only DAPD*

*would explore a strong overshoot and net negative emission behavior and without IAPD it would be impossible to adequately illustrate the lower region of plausible futures. "*

There is only one scenario with a substantial overshoot behaviour (DAPD). From an earth system perspective, I would expect significant value in modeling overshoot also at higher warming levels (exceeding and declining below 2 degrees), over long time scales, and with more substantial overshoot (simply to deal with noise). This could lose realism or political palatability of such a pathway, but it is an example where I would appreciate a clearer discussion of how different considerations in the science and policy space might lead to different sets of REPs, so that the authors can then transparently disclose their judgement in their final selection.

REPLY1.27c: This is indeed a relevant point. Yet also one that is directly linked to the question of timescales. CMIP6 scenarios are designed until 2100, we'd suggest to design REPs by default until 2150. But this will be barely enough to have a plausible "substantial overshoot" pathway (for example going to 2.5°C in 2100 and then back down). Which means there are two options to implement something along the lines of what the reviewer suggests. The first one is to generate hypothetical extreme emission scenarios, such as as arguably SSP5-34-OS was. Those more hypothetical scenarios to learn about system behaviour are indeed beneficial but lack the direct policy relevance.CDRMIP during CMIP6 explored those hypothetical scenarios well and as mentioned in now further above in the manuscript, the reader ought to consider these framework scenarios in conjunction with the scientifically-oriented specialised MIPs. Learning about hypothetical futures can easily (and arguably) better happen with clearly defined, possible single gas extreme case scenarios. We argue that priority should be given to the policy-relevant overshoot magnitudes and timings within the scope of the framework scenarios, which is why we put forward the DAPD (delayed action peak and decline) scenario.

The second, and maybe more meaningful option, would be to use the scenario extensions beyond 2150 to explore different overshoot futures. I.e. design a NFA extension between 2150-2500 that returns from 2.5-3°C back to 1.5°C (or even lower) on multi-century timescales. In our manuscript, we do not discuss the extensions in great detail, but have added this suggestion now explicitly in the table 1 when discussing the NFA scenario:

*"A set of further extensions of a pathway from this category beyond 2150 would be helpful for investigating tipping elements or slow response in the Earth system such as ice sheet **or permafrost, as well as long-term temperature decline, very high overshoot pathways."***

We also added the following paragraph just before the discussion section:

*"We note that for some of the research questions identified, in particular in relation to overshoot and long-term (ir)reversibility, the pathway extensions beyond 2150 are of particular relevance. Stylised extensions for SSP/RCP scenarios have been provided until 2500 and we suggest considering this timeframe also for extending the framing pathways. However, we would suggest moving beyond stylised extensions and explicitly consider the policy and research questions we outlined above in the design of the pathway extensions. In particular, it might be advisable to consider more than one extension per pathway i.e. explore the effects of a long-term temperature stabilisation vs. decline from the same emission pathway in 2150 (Lamboll et al. 2022). "*

RC1.28: L617ff: This section is critical for the argument whether the set of REPs that is proposed for a broader narrative framing purpose necessarily needs to be run, in its entirety, through ESMs to address the questions that motivated the selection of REPs in the first place. I don't think this section quite does justice to this need. I find L628-633 unclear and unconvincing: are the authors saying that we could rely on only small ensembles per model, but a wider set of models, to address signal-to-noise in close-by REPs? How does this approach handle the fact that CMIP contributions are still an ensemble of opportunity rather than necessarily providing robust statistics? Also in L631-636, regional differences and differences related to land-use may indeed be much larger than the global mean, but if we have only one or two REPs it will be very difficult to learn how much a regional difference is due to models or a reflection of the particular SLCF and land-use emission choices made within REPs. L645/646 argues that ESM runs of overshoot pathways are necessary to enable WGII to engage with impacts and risks under overshoot – I'm not sure this is the case, as the main questions that WGII needs to address to better understand risks under overshoot (but, importantly, also the avoided risks once temperatures start declining again) in my view are more fundamental about process understanding, reversibility and cumulative damages, not necessarily whether we have highly resolved ESM outputs to drive impact models. Lastly, as flagged earlier, the section leaves open whether we learn as much as we need to from those scenarios about earth system behaviour in

response to net negative emissions, or whether there is a scientific case for a more extreme overshoot scenario. Such a scenario would be politically unpalatable and this may limit its value as a narrative 'framing' scenario, but it might provide critical knowledge from ESMs about processes and thresholds. Is that the case? If yes, how would the authors propose to resolve this tension?

REPLY1.28: Again, we are thankful to the reviewer for raising these pertinent points. On the last point, as we briefly mentioned, this would be best addressed either exploring more hypothetical scenarios housed in the specialised MIPs, which can provide dedicated scenario protocols to test and scientifically understand model behaviour under more 'clean' experimental settings, or as 2500 scenario extensions.

The questions of identifying causal drivers such as SLCFs and land-use is a pertinent one that we'd hope can be explored with variations of the REPs in specialized MIPs.

With regard to the point of how proposed design of the framework scenarios deals with the issue that participation of models in CMIP does not represent a random sample, but a hard-to-interpret collection of interdependent models with different ensemble sizes (ensemble of opportunity): We do not attempt to solve that issue, as it is intricate and going beyond the issue of framework scenario design. Possibly, new weighting techniques can in the future continue that path that IPCC AR6 WG1 started (see Chapter 4), i.e. that some performance criteria in terms of observational constraints are employed to construct the range of projected climate change under a particular scenario. But again, we consider that to be outside the scope of our paper. The only reason we brought up the ensembles of individual models is that some models tend to be run in large ensemble settings which will enable insights on the attributable differences between close-by scenarios that individual ensemble runs cannot. Yet having small (noisy) ensembles of these differences across the full ESM range at the same time will enable a quantification of climate change due to 'close-by' scenarios that has not been possible previously over the 21st century.

RC1.29: L695ff: The focus on regional downscaling misses the broader question of whether a ESM run produced with a close-by REP could be scaled to infer regionally relevant information based on a more finite set of REPs. There is also the question whether other, more specialised models need separate ESM runs for close-by REPs, or whether they could be driven with a smaller set of ESM runs plus other information, and/or scaled ESM results from close-by REPs to approximate climate drivers for e.g. ice sheet models. In an ideal world, we would of course model all REPs through ESMs, but given resource constraints, are there short-cuts that would be valid to take – and are there short-cuts that would be a scientific mistake? (Or a political/narrative mistake?)

REPLY1.29: Important question, yet we first want to clarify that section 695 is NOT about regional *downscaling*. 'Regional downscaling' is indeed a different field which we consider to

be a different one and outside of the scope of this paper. Downscaling refers to whether a coarse resolution ESM can be either dynamically (with regional climate models) or statistically (with the use of observational data) downscaled to finer resolutions in areas of interest. Irrespective of the choice of framing scenarios, that remains a valid, but separate, area of research. We here refer to the regionally explicit emulators which intend to either simply 'scale' ESM outputs from one scenario to the other or employ different re-arrangement techniques or even machine-learning approaches to emulate potential ESM outcomes for one scenario from the insights learned from others. As we argue, and despite the fact that multiple of the authors are also strongly involved in these kinds of emerging approaches, we argue that they are not yet able to replace the inherently consistent ESM output for most of the impact studies. The same discussion arises almost in every IPCC assessment cycle. For example, pattern scaling approaches were argued at the time of IPCC AR4 to fill in for lower scenarios, as AOGCMs at the time were asked to run a 'constant concentration' scenario apart from higher SRES scenarios. Apart from the scientific reasons on the limitation of these approaches to date, the mere empirical evidence over the IPCC AR4, AR5 and AR6 cycle confirm that the combination of scientific shortcomings, a lack of a 'standardisation' to allow integration of results across multiple WG1, WG2 communities and less 'standing' throughout both the scientific and political communities means that (pattern-)scaled scenarios are unlikely to ever yield the same prominence, ability to integrate communities and policy-relevant high level IPCC insights as scenarios that were directly examined by the world-class suite of ESM models.

RC1.30: L783-795: I don't disagree with the points being made, but I don't think the manuscript has quite made the case that these needs are best and can only be addressed by running all those REPs through ESMs. What REPs best support framing of alternative (and foregone) futures under Paris and in the IPCC is not necessarily the same set as what REPs should be used to drive ESMs to learn what we need to and can only learn from ESMs. A more critical reflection on whether there are differences, and if so, whether those differences must be overcome to nonetheless end up with a single set, or whether there could be different sets or subsets of REPs that serve different purposes, would in my view lift the value of this (already highly valuable) manuscript.

REPLY1.30: Thank you very much. Please see our response to this point above under REPLY1.2. In addition, we agree fully with the reviewer that additional scenarios are of use, both to be run by ESMs but also to complement the ESM-REPs for (emulator-evaluated) scenarios. As this paper however focuses on the set of scenarios to be run by ESMs, it would in our opinion stretch the scope of the paper too much if we present - apart from a general framework - also detailed discussions on the useful REPs that are not to be run by ESMs.

TECHNICAL CORRECTIONS

Citations of Pirani et al (submitted), and Sanderson et al (in prep) should presumably not appear in the final version of this manuscript, but only published literature?

REPLY: Thank you. Yes, these references will be adapted to

Sanderson (2023) as per: https://egusphere.copernicus.org/preprints/2023/egusphere-2023-2127/

And Pirani (2024) as per: https://www.nature.com/articles/s44168-023-00082-1

Citations to IPCC reports, and Glossary: please follow the IPCC guidance on how to cite those reports. IPCC reports as a whole should not be cited by first author et al, but as IPCC (with an editorial team). Also see guidance on how to cite the glossary.

REPLY: Thank you. We will adapt these IPCC references.

Fix citation of WMO (currently it is World Meteorological, O.)

REPLY: Will be done. Thanks.

Please provide full citation details for the ScenarioMIP report (van Vuuren et al 2023)

REPLY: Thank you. Will be done.

L239: stray "e" before "REPs"

REPLY: Corrected.

L339: suggest adding a paragraph mark to differentiate the discussion of high-end scenarios from the following discussion of a low-end scenario.

REPLY: Thank you. Implemented. We also adapted the previous text to an easier-to-read list.

L380: the word "as" appears to be missing after "insofar"

REPLY: Implemented.

L454: space missing in "consequencesincluding"

REPLY: Corrected.

L727: "comp" -> "compounds" (?)

REPLY: Thank you. Corrected.

L805: can't make sense of grammar here: "… in this critical and contribute to…"

REPLY: Thank you for spotting that glitch. We simply deleted 'in this critical'.

References:

Asaadi, A., Schwinger, J., Lee, H., Tjiputra, J., Arora, V., Séférian, R., Liddicoat, S., Hajima, T., Santana-Falcón, Y., and Jones, C. D.: Carbon cycle feedbacks in an idealized simulation and a scenario simulation of negative emissions in CMIP6 Earth system models, Biogeosciences, 21, 411–435, https://doi.org/10.5194/bg-21-411-2024, 2024.

Pfleiderer, Peter, Carl-Friedrich Schleussner, and Jana Sillmann. "Limited reversal of regional climate signals in overshoot scenarios." *Environmental Research: Climate* 3, no. 1 (2024): 015005.

Santana-Falcón, Y., Yamamoto, A., Lenton, A. et al. Irreversible loss in marine ecosystem habitability after a temperature overshoot. Commun Earth Environ 4, 343 (2023). https://doi.org/10.1038/s43247-023-01002-1

Schwinger, Jörg, Ali Asaadi, Norman Julius Steinert, and Hanna Lee. "Emit now, mitigate later? Earth system reversibility under overshoots of different magnitudes and durations." *Earth System Dynamics* 13, no. 4 (2022): 1641-1665.

**Author replies to RC2**

We thank the reviewer Dale S. Rothman for his in-depth approach to reviewing our contribution. Please find our replies in bold below.

**Meinshausen et al: A perspective on the next generation of Earth system model scenarios: towards representative emission pathways (REPs)**

**Review by Dale S. Rothman**

**Preface**

In preparing this review, I first went back and reviewed the SPMs from IPCC AR6, the report of the IPCC Workshop on the Use of Scenarios in the Sixth Assessment Report (Masson-DelmoFe et al 2023), and the report of ScenarioMIP Workshop: Pathway to Next Generation Scenarios for CMIP7 (van Vuuren et al 2023). Furthermore, I considered the proposal for Representative Emission Pathways (REPs) in the broader context of scenario use in climate change analysis. Specifically, I have adopted the position that evaluating a proposal for a set of representative scenarios needs to keep in mind their potential use for various types of analysis. This points to the importance of being able to interpolate (not extrapolate), both spatially and temporally, the information provided by the representative scenarios in order to meet the needs of these other analyses. I have sub-divided my comments into three parts: 1) Issues with Representation of the RCP-SSP-SPA Framework, 2) Choice to use Emissions Pathways, and 3) Specific Emissions Pathways Recommended.

Reply RC2.1: Much appreciated.

**Issues with Representation of the RCP-SSP-SPA Framework in AR6**

My comments here focus on a number of statements that I feel mis-represent the RCP-SSP-SPA framework. Specifically, the statements "the so-called SSP-RCP matrix (Moss et al., 2010; van Vuuren et al., 2014) was used to explicitly present the climate and socio-economic dimensions as independent dimensions" and "Shared Policy Assumptions (SPAs) were used to vary the climate outcomes (Kriegler et al., 2014)" on p. 6 concern me. While these issues are less important to the present paper as my later issues, they do need to be recognized.

The RCPs were developed early on as a quick way to provide quantitative emission/concentration pathways as inputs to ESMs and GCMs. The nomenclature, e.g., RCP8.5,

was, unfortunately a bit confusing as 8.5 referred to the associated radiative forcing estimated in the IAMs rather than either the levels of emissions/concentrations. The pathways did have underlying socio-economic and land-use assumptions as inputs into the IAMs, which needed them to produce the emission/concentration pathways, but these were not intended to be examined in any great detail.

The SSPs were subsequently developed by thinking more carefully and consistently about the underlying socio-economic aspects, e.g., population, education, economic, and technological changes. This did help to produce some key quantitative results for each SSP using, among other things economic and education models, but also some more qualitative elements[2]. These were used as inputs to the IAMs, which were used to produce, among other things, new quantitative pathways of emissions, concentrations, and radiative forcing, also to be used as inputs to ESMs and (AO)GCMs. The initial SSPs excluded considerations of explicit climate policy, but later versions were developed to produce pathways with lower levels of radiative forcing. Although some studies depicted as using, e.g., SSP1-RCP1.9, did combine the socio-economic elements from an SSP with the emissions/concentrations/radiative forcing pathways from one of the original RCPs, the intention was that studies should actually use the emissions/concentrations/radiative forcing pathways produced from running the outputs of the IAM run with the assumptions from the SSP.

Finally, the SPAs were meant to reflect the fact that different climate policies made more sense in different SSPs and, therefore, make the SSPs including climate policies more internally consistent. For example, global participation did not really fit well with the story being told in SSP4. While this did lead to differences in the climate outcomes, it is inaccurate to say that this was their purpose.

Reply RC2.2: We much appreciate this much more encompassing reflection on SSPs, RCPs and SPAs which we fully agree with (as some of us have been strongly involved in some of its aspects). This reflection is really a nice balanced view. And we also correct the well-spotted inaccuracy in our previous very succinct wording, so that the corrected statement now reads:

> *"Shared Policy Assumptions (SPAs) were employed to represent diverse policy assumptions, which led to varying emission levels for the same SSPs (Kriegler et al., 2014)."*

> Instead of:

*"Shared Policy Assumptions (SPAs) were used to vary the climate outcomes (Kriegler et al., 2014)."*
* * *
[2] The qualitative elements were important for the SPAs as noted in the next paragraph, but also for further, usually IAV, studies that needed more details than provided by the quantitative information.

**Choice to use Emissions Pathways**

Meinshausen et al choose to focus on Representative Emission Pathways rather than pathways identified by either concentrations or levels of radiative forcing. They discuss some pros and cons of this choice already, but I would like to add a few additional comments.

On the pro side, using emission pathways would likely lead to 'cleaner' pathways coming out of the IAMs, which they note are "likely to be derived from IAMs". This assumes that the IAMs are able to provide these emission pathways prior to any changes resulting from their own internal carbon cycle and climate components and resulting feedbacks. Otherwise, there could be issues of conflicting effects with the carbon cycle and climate components and resulting feedbacks in the ESMs and (AO)GCMs.

Reply RC2.3: Yes, we agree. Although this issue of potential conflict between the IAM's 'internal carbon cycle and climate components and resulting feedbacks' and the ESMs and (AO) GCMs has historically been somewhat addressed by using a single calibrated emulator across all the IAMs that translates emissions to the concentrations - thereby at least avoiding a situation where different kinds of feedback parameterisations affect each scenario differently. Anyway, but yes, we agree with this argument. See also Sanderson et al. for a deeper discussion with regards to the pros and cons of CO2 emission driven scenarios (https://egusphere.copernicus.org/preprints/2023/egusphere-2023-2127/ ).

On the con side, there are several issues, two of which I note here:

• Any single REP would likely be associated with any number of concentration, warming, hazard, and impact and risks pathways. The first three of these relate to the models used to estimate concentrations, warming, and hazards. As for the impact and risk pathways, these can differ given that it is likely that there could be a many-to-one relationship between socio-economic/policy pathways and emission pathways, as the former are important considerations in estimating impacts and risk.

Reply RC2.4: Yes, we very much agree again. However, we do not see a strong difference between concentration-driven and emission-driven runs in this respect. Already under the concentration-driven runs in the past, different ESMs produced different warming futures

and geophysical hazards. The one additional uncertainty that will get reflected in the warming futures and hazards will be the (carbon cycle) uncertainties. And that is probably a very good thing in terms of appropriately enabling risk frameworks to deal with a more full set of uncertainties rather than a strongly truncated one. The fact that there is a one-to-many mapping between an emissions pathway and e.g. impacts is something we want to make explicit here (and it seems to be under-appreciated in many key applications).

- The IAMs are limited in their spatial resolution, with some elements represented at national/regional level and others at grid cell level. The former becomes an issue when considering, for example, non-CO2 GHG (including particulates) emissions and certain impacts. Some of this might be ameliorated with good downscaling of these elements

*Reply RC2.5: Again, we fully agree with this analysis. On the other hand, a concentration driven scenario does not resolve these spatial uncertainties but somewhat circumvents them. In terms of the short-lived forcers, in particular aerosols, ESMs have already been driven with aerosol emissions from IAMs in the past and dedicated downscaling techniques have been developed and applied to provide those. Thus, there would not be a change of approach in terms of aerosols / particulates\*. Arguably the only difference that we suggest in this regard is for carbon dioxide emissions (which should be used as input instead of concentrations), as many of the ESMs will also in this coming CMIP7 cycle be dependent on non-CO2 greenhouse gases being provided as concentration fields.*

**Specific Emissions Pathways Recommended**

The specific REPs proposed are reasonable, but I do have a couple of concerns.

- In describing the pathways, e.g., in Table 1, the authors need to be clear in the names and the key characteristics that they are meant to be illustrative of ways that the emission pathways might come about. Specifically, the current names and descriptions of the key characteristics include explicit assumptions about socio-economic, technological, and policy elements. This is at odds with their desire that "the REPs should remain separated from the underlying socio-economic

scenarios."

*Reply RC2.6: Thank you for this suggestion. Our explicit assumptions are meant to only provide large scale definitions of the runs, such as "current policy" implementations. In what way the IAMs extend such a "current policy" setup in the future is obviously subject to a range of explicit and implicit assumption within each IAM modelling framework and socio-economic input assumptions. We indeed do not intend to prescribe the latter.*

- None of the pathways deal with the possibility of the use of SRM, which would present some of the same, but at the sametime quite different, challenges for ESMs and (AO)GCMs as pathways with net negative emissions at some point in the future. It has been argued elsewhere that SRM is not currently part of the climate change policy debate, but this is increasingly inaccurate and likely to be even more so over the time frame over which these REPs are meant to be applied. Not including some consideration of SRM would set up the community to be in a position of doing a lot of catching up in the future, much as has been argued about the need to do a better job of dealing with CDR at this point.

*Reply RC2.7: Again, this is an important point. We would argue in this paper that SRM approaches are less policy-relevant as of now and are best researched in a continuation of the successful GeoMIP6 endeavor rather than within one of the framework scenarios. The framework scenarios could serve as reference scenarios also for a future GeoMIP comparison protocol (Kravitz et al., 2015). Thus, the GeoMIP participating ESMs would run the respective framework scenario, e.g. DASMT, and then also apply a sensitivity run. We included an explicit reference to GeoMIP in our list of specialised MIPs in section 3, where it now says "Thus, many of the scientific advances can be expected to emanate from the specialised MIPs (CFMIP, HighResMIP, AerChemMIP, C4MIP, RFMIP, CDRMIP, GeoMIP, LUMIP, ISMIP, OMIP, VolcMIP, DAMIP etc - see https://wcrp-cmip.org/model-intercomparison-projects-mips/) that are conducted in parallel to running ESMs with the multi-gas scenarios."*

**A Final Comment**
I enjoyed reading this paper and feel that it is an important contribution to the further development of scenarios for CMIP7 and AR7. I look forward to the final version.

*Reply RC2.8: We very much appreciate the time and insights provided in this review.*

**References**

Masson-Delmotte, V., Pörtner, H.-O., Roberts, D. C., Shukla, P. R., Skea, J., Zhai, P., Cheung, W., Fuglestvedt, J., Garg, A., O'Neill, B., Pereira, J., Portugal Pereira, J., Riahi, K., Sörensson, A., Tebaldi, C., ToOn, E., van Vuuren, D., Zommers, Z., Al Khourdajie, A., Connors, S. L., Fradera, R., Ludden, C., McCollum, D., Mintenbeck, K., Pathak, M., Pirani, A., Poloczanska, E. S., Some, S., and Tignor, M.: Workshop Report of the Intergovernmental Panel on Climate Change Workshop on the Use of Scenarios in the Sixth Assessment Report and Subsequent Assessments, Working Group III Technical Support Unit, Imperial College London, United Kingdom, 67, 2023.

van Vuuren, D., Tebaldi, C., O'Neill, B. C., SSC, S., and participants, w.: Pathways to next generation scenarios for CMIP7: ScenarioMIP workshop report. 2023.